

# Estimation of turbulence parameters from scanning lidars and in-situ instrumentation in the Perdigão 2017 campaign

Norman Wildmann[1], Nicola Bodini[2], Julie K. Lundquist[2,3], Ludovic Bariteau[4], and Johannes Wagner[1]

[1]Deutsches Zentrum für Luft- und Raumfahrt e.V., Institut für Physik der Atmosphäre, Oberpfaffenhofen, Germany
[2]Department of Atmospheric and Oceanic Sciences, University of Colorado Boulder, Boulder, Colorado, USA
[3]National Renewable Energy Laboratory, Golden, Colorado, USA
[4]Cooperative Institute for Research in the Environmental Sciences, University of Colorado Boulder, Boulder, Colorado, USA

**Correspondence:** Norman Wildmann (norman.wildmann@dlr.de)

**Abstract.** The understanding of the sources, spatial distribution and temporal variability of turbulence in the atmospheric boundary layer (ABL) and improved simulation of its forcing processes require observations in a broad range of terrain types and atmospheric conditions. In this study, we estimate turbulence kinetic energy (TKE) dissipation rate $\varepsilon$ using multiple techniques, including traditional in-situ measurements of sonic anemometers on meteorological towers, a hot-wire anemometer on a tethered lifting system (TLS), as well as remote-sensing retrievals from a vertically staring lidar and two lidars performing range-height indicator (RHI) scans. For the retrieval of $\varepsilon$ from the lidar RHI scans, we introduce a modification of the Doppler Spectral Width (DSW) method. This method uses spatio-temporal averages of the variance of the line-of-sight (LOS) velocity and the turbulent broadening of the Doppler backscatter spectrum. We validate this method against the observations from the other instruments, also including uncertainty estimations for each method. The synthesis of the results from all instruments enables a detailed analysis of the spatial and temporal variability of $\varepsilon$ across a valley between two parallel ridges at the Perdigão 2017 campaign. We find that the shear zones above and below nighttime low-level jets (LLJ) experience turbulence enhancements, as does the wake of a wind turbine (WT). We analyze in detail how $\varepsilon$ varies in the early morning of 14 June 2017, when the turbulence in the valley, approximately eleven rotor diameters downstream of the WT, is still significantly enhanced by the WT wake.

## 1 Introduction

While turbulence is the major driving force for mixing in the atmospheric boundary layer (ABL), its variability introduces challenges for modelling turbulent processes. In heterogeneous, complex terrain, turbulence can be triggered or enhanced by many different flow features, such as recirculation or detachment caused by mountains (Stull, 1988; Menke et al., 2019) or blockage at the edges of forests or other obstacles (Irvine et al., 1997; Dupont and Brunet, 2009; Mann and Dellwik, 2014). In highly complex terrain sites, forests or patches of trees with varying canopy density and height induce variable mixing processes (Belcher et al., 2012). In stable atmospheric conditions, wave-like motions cause intermittent turbulence which is only poorly understood (Sun et al., 2015). Convection and thermally driven flows pose an additional challenge (Adler and Kalthoff, 2014). In contrast to natural sources of turbulence, wind turbines generate vortices at the rotor blades which propagate downstream





and disperse in a wake, a region of high turbulence which interacts with the surrounding atmosphere (Lundquist and Bariteau, 2015). Due to these diverse phenomena, parametrization of the turbulent mixing and drag that is caused in such environments is highly challenging for numerical simulations and can cause significant errors.

In forecasting models, the ABL is parameterized by a set of equations that describes the turbulent mixing (Nakanishi and Niino, 2006). Goger et al. (2018) found that in the COSMO-model, turbulence kinetic energy (TKE) is systematically underestimated with a one-dimenstional turbulence parameterization. Recent sensitivity studies by Yang et al. (2017) showed that the parameters associated with turbulent mixing in an ABL parametrization have a large impact on 80 m wind speeds in the Weather Research and Forecasting model (WRF). They find that parameters associated with turbulence dissipation rate are responsible for approximately 50% of the variance of 80 m wind speeds. Muñoz-Esparza et al. (2018) used turbulence measurements from sonic anemometers at the XPIA campaign (Lundquist et al., 2017) to motivate improvements in WRF boundary-layer parametrizations.

While observations are essential to improve our understanding and simulation of turbulent processes, the retrieval of turbulence parameters from measurements is not trivial, especially at complex sites. Sonic anemometers are a reliable tool to resolve the small scales of turbulence, allowing the calculation of turbulence parameters at fixed points in space (Champagne, 1978; Oncley et al., 1996; Beyrich et al., 2006). However, point measurements are not necessarily representative of turbulent mixing in a larger area. Recent developments in commercial scanning lidars can provide an assessment of turbulent mixing over a broader region (Smalikho et al., 2013), and so many different methods have been introduced to retrieve vertical profiles of turbulence from either vertical stare measurements (O'Connor et al., 2010; Bodini et al., 2018, 2019), six-beam scanning scenarios (Sathe et al., 2015; Bonin et al., 2017) or vertical azimuth display scans (VAD Eberhard et al., 1989; Smalikho and Banakh, 2017). Krishnamurthy et al. (2011) derived vertical profiles of TKE from horizontal plan-position indicator (PPI) scans. Using more than one lidar, multi-Doppler retrievals of the three-dimensional wind vector are possible and the obtained wind data can be analyzed for turbulence parameters (Newsom et al., 2008; Röhner and Träumner, 2013; Iungo and Porté-Agel, 2014; Pauscher et al., 2016; Wildmann et al., 2018b).

Here, we demonstrate a new approach for assessing the variability of turbulence parameters in complex terrain by employing multiple instruments to provide a comprehensive view of turbulence structures and variability at the Perdigão 2017 field campaign (details in Sect. 2). A new method is introduced to retrieve turbulence TKE dissipation rate $\varepsilon$ from lidar range-height indicator (RHI) scans, which allow a two-dimensional perspective of the turbulence in the valley between two ridges. These retrievals are calibrated with data from sonic anemometers on meteorological towers and validated with more established measurements of $\varepsilon$ from lidar vertical stares and high-resolution hot-wire anemometer measurements on a tethered lifting system (TLS). The different approaches to retrieve TKE dissipation rates are explained in Sect. 3 and results of the validation are given in Sect. 4, highlighting the limits where TKE dissipation can be quantified. In Sect. 5, a case study is presented, in which wind-turbine-induced (wake) turbulence is detected by measurements in the valley under specific atmospheric conditions. Prospects for future refinement of these methods and inclusion of these new insights into model parameterizations are highlighted in the Conclusions.





## 2 Experiment description

### 2.1 The site

Perdigão is a village in central Portugal, approximately 25 km West of Castelo Branco and eponymous for an international
field campaign with the goal of studying the microscale flow over two nearly parallel mountain ridges. The two mountain
ridges at Perdigão are oriented approximately 35° from North in the counter-clockwise direction, running from northwest to
southeast. Figure 1 shows a map of the experimental site, rotated by 35° and focusing on the Vale do Cobrão in between
the two mountain ridges. According to long-term measurements before the field campaign, the primary wind direction at the
site is south-westerly, perpendicular to the ridge orientation (Fernando et al., 2019). A secondary wind pattern, that mainly
occurs in nighttime, is north-easterly flow, also perpendicular to the ridges. Wagner et al. (2019a) provides a detailed analysis
of the meteorological situation during the period of intensive operation (IOP) of the campaign from 1 May to 15 June 2017.
To visualize the complexity of the site not only in terms of the topography, but also in terms of land use, roughness elements
derived from a high-resolution aerial laser scan have been added to the map and show the patchwork of small areas of trees
and forest. The vegetation data was collected in March 2016, approximately one year before the campaign, so because of rapid
vegetation growth, these data do not exactly represent vegetation heights during the IOP. A picture showing an aerial view of
the site during the campaign can be seen in Fig. 2.

After initial pre-studies at the site (see Vasiljević et al., 2017), the 2017 campaign brought together a unique amount of re-
searchers interested in the microscale of complex terrain flows. A comprehensive description of the scientific goals of all
contributing partners, as well as an overview of the instrumentation installed in the campaign can be found in Fernando et al.
(2019). More than 180 sonic anemometers on 49 meteorological masts and 26 lidar systems of different kinds were installed
to sample the complex three-dimensional flows in the valley between the mountain ridges as well as in the inflow and outflow
regions.

### 2.2 Instrumentation

#### 2.2.1 Scanning lidars

Here, we focus on the flow in the center of the valley and in a cross-section through the location of the wind turbine. For
this purpose, three Leosphere Windcube 200S lidars of the German Aerospace Center (DLR) were deployed at the locations
indicated in Fig. 1. All of the systems performed RHI-scans as indicated by the dashed lines in Fig. 1. Lidar #1 and #2 were
aligned with the wind-turbine along the primary wind direction. Lidar #2, in the valley, probed the flow with a high elevation
angle, so that the line-of-sight (LOS) measurements included a significant contribution from the vertical wind component.
Lidar #1, on the northeast ridge, probed the valley flow at a low elevation angle, thus measuring primarily contributions of
the horizontal wind components. Synthesizing data from these two lidars allows coplanar wind speed retrievals as described
in Wildmann et al. (2018a). Lidar #3 provided additional information about the out-of-plane flow and WT wake position. A
combination of the three lidars also allowed flexible multi-Doppler measurements of the wind turbine wake for a range of wind



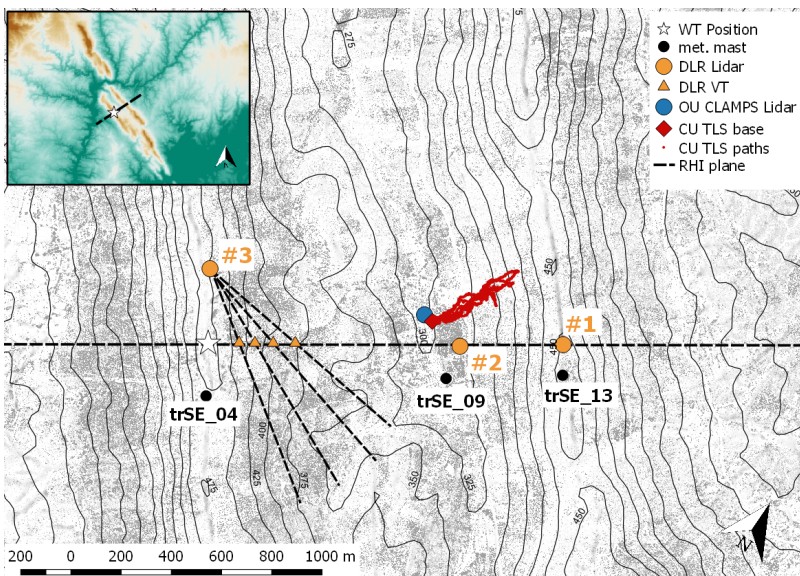

**Figure 1.** Map of the Vale do Cobrão. The grey structures are surface elements (mostly forest) obtained from a high resolution lidar elevation scan one year before the campaign. The map is rotated to the orientation of the ridges. The small map in the top-left gives a wider overview of the surrounding area. The dashed lines show the cross-sections for measurements with the lidar instruments used in this study.

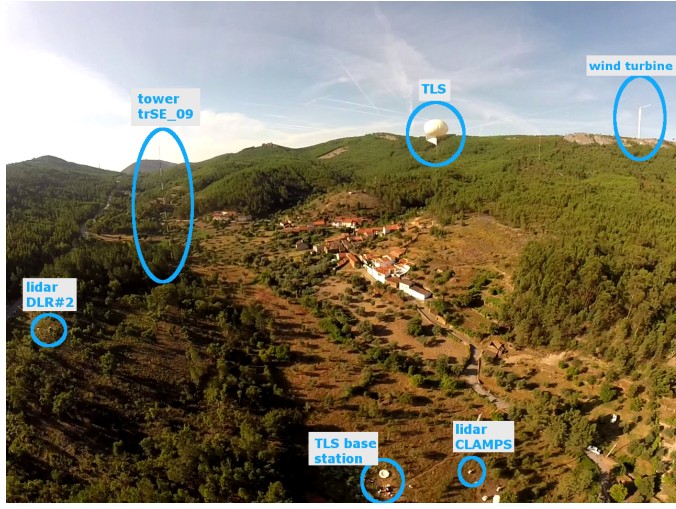

**Figure 2.** Aerial view over the measurement site from North.



**Table 1.** Main technical specifications of the Leosphere Windcube 200S lidars.

|  | DLR#1 | DLR#2 | DLR#3 |
|---|---|---|---|
| Wavelength | 1.54 µm | | |
| Scan speed | $2°\,\mathrm{s}^{-1}$ | | $1°\,\mathrm{s}^{-1}$ |
| Accumulation time | 500 ms | | 1000 ms |
| Angular resolution | $1°$ | | |
| Azimuth | 237° | | 95°,105°,115°,125° |
| Range gate distance ($\Delta R$) | 20 m | 10 m | 10 m |
| Specified pulse length | 200 ns | 200 ns | 200 ns |
| Scan duration | 51 s | 77 s | 52 s |
| Min. elevation | $-12°$ | $6°$ | $-2°$ |
| Max. elevation | $90°$ | $160°$ | $50°$ |

**Table 2.** Main technical specifications of the Halo Streamline lidar.

| | |
|---|---|
| Wavelength | $1.548\,\mu\mathrm{m}$ |
| Receiver bandwidth | $\pm 25\,\mathrm{MHz}$ |
| Nyquist velocity ($B$) | $\pm 19.4\,\mathrm{m\,s}^{-1}$ |
| Signal spectral width ($\Delta\nu$) | $2\,\mathrm{m\,s}^{-1}$ |
| Pulses averaged ($n$) | 20000 |
| Points per range gate ($M$) | 10 |
| Time resolution | $\sim 1\,\mathrm{Hz}$ |
| Elevation angle | $90°$ |

directions far from the main wind direction (Wildmann et al., 2018b). The parameters of the RHI scans and lidar specification are given in Tab. 1.

As part of the Collaborative Lower Atmospheric Mobile Profiling System (CLAMPS, Wagner et al., 2019b), the University of Oklahoma (OU) deployed a Halo Photonics Streamline Scanning lidar at the so-called Lower Orange Site in the Vale do Cobrão, approximately 100 m from the cross-section through the WT. The scanning scenarios for this lidar during the campaign comprised a regular series of velocity azimuth display (VAD) scans, RHI along-valley and cross-valley scans, and vertical stare measurements. In this study, the vertical stare measurements are used to derive turbulence dissipation rate and the results from the VAD-scan is used for wind speed information. Table 2 gives an overview of the OU CLAMPS lidar parameters for the vertical stare measurements that are relevant for the turbulence retrieval (see Sect. 3).



### 2.2.2 Tethered lifting system

The University of Colorado Boulder's tethered lifting system (TLS), a specialty-designed tethersonde system, enables unique *in situ* high-rate measurements of wind speed, wind direction, and temperature. From these high-rate measurements, TKE dissipation rate can be estimated (Frehlich et al., 2003, 2008). TLS capabilities for observing detailed wind speed, temperature, and dissipation rate profiles have been demonstrated in several field campaigns (Balsley et al., 2003; Frehlich et al., 2008), including measurements of dissipation rate in wind turbine wakes (Lundquist and Bariteau, 2015). Muschinski et al. (2004) used data from the TLS to assess small-scale and large-scale turbulence intermittency in flat terrain, while Sorbjan and Balsley (2008) used the system to explore microscale turbulence in the stable boundary layer.

The Perdigão TLS instrument packages were similar to those of Frehlich et al. (2008). Fast wind speed measurements at 1 kHz were from 1.25 mm length, 5 micron diameter Tungsten wires. Other measurements included 1 kHz coldwire anemometer temperature measurements (Auspex Scientific, custom-made), 100 Hz thermistors (Honeywell 111-103EAJ-H01), solid-state measurements for temperature (Analog Devices Inc TMP36) and relative humidity (Honeywell HIH-4000), a 100 Hz Pitot tube (Dwyer instruments model 166-6) and pressure sensor (Honeywell DC001NDC4) for velocity and pressure measurements, as well as GPS and compass measurements. GPS measurements of latitude, longitude, and altitude were sampled every 5 s. While the TLS can be deployed in either a profiling or a hovering mode, most Perdigão measurement consisted of profiles. The ascent and descent of the TLS was controlled by a custom-made winch system with an average ascent/descent rate of $0.3 \mathrm{~m\,s^{-1}}$. The payloads were lifted using a 16 $\mathrm{m^3}$ helikite system (Allsop) with a lightweight but strong tether (1120 kg Dyneema line 2.5 mm diameter).

### 2.2.3 Sonic anemometers on meteorological masts

Out of the over 180 sonic anemometers on more than 40 meteorological masts, this study relies on the instruments on towers 20/trSE04 and 25/trSE09. Both of these towers were 100 m high with sonic anemometers at 10 m, 20 m, 30 m, 40 m, 60 m, 80 m and 100 m levels on booms pointed to ~135° (~155°) for tower 20/trSE04 (25/trSE09). All the sonics on these two masts were Gill WM Pro sonic anemometers, sampling the three-dimensional wind vector at a rate of 20 Hz. The two sonic anemometers at 80 m and 100 m at the 100 m meteorological mast 25/trSE09 were within the height limits of the DLR and OU lidar scans and are therefore used for intercomparison of turbulence measurements. The ground levels of tower 20/trSE04 and 25/trSE09 with respect to ridge height (i.e. wind turbine base height) are $-10\ m$ and $-178\ m$ respectively.

## 3 Methods

### 3.1 Basic equations and terminology

The quantification of turbulence from measured data in boundary-layer meteorology is often based on the assumption of homogeneity and local isotropy in the small scales of turbulence which has been found valid in high Reynolds-number flows (Kolmogorov, 1941). Under these assumptions, the energy cascade of eddies from larger to smaller scales in the inertial sub-



range of turbulence can be defined by a model for the energy spectral density $S(\kappa)$:

$$S(\kappa) = \alpha \varepsilon^{2/3} \kappa^{-5/3} \quad , \tag{1}$$

where $\kappa$ is the wavenumber, $\varepsilon$ is the TKE dissipation rate and $\alpha$ is a universal constant. Integration of the energy spectrum yields the variance $\sigma^2$:

$$\sigma^2 = \int\limits_{-\infty}^{\infty} d\kappa S(\kappa) \quad . \tag{2}$$

Turbulence of the velocity field can be described by the structure function $D$, which can be calculated from flow velocities $v$ as the square of differences at spatially separated points:

$$D_v(r) = \langle [v(x+r) - v(x)]^2 \rangle \quad , \tag{3}$$

where $r$ is the separation distance and $\langle \rangle$ is used to symbolize the ensemble average. By invoking Taylor's hypothesis of frozen turbulence, a separation distance can be converted to a separation time in a homogeneous flow with a mean flow velocity $\overline{v}$, so that $\tau = \frac{r}{\overline{v}}$ and

$$D_v(\tau) = \langle [v(t+\tau) - v(t)]^2 \rangle \quad . \tag{4}$$

Kolmogorov (1941) formulated that the structure function scales with dissipation rate $\varepsilon$ and the Kolmogorov constant $C_k \approx 2$ according to

$$D_v(r) = C_k \varepsilon^{2/3} r^{2/3} \quad . \tag{5}$$

Smalikho et al. (2005) derives the longitudinal spectrum of flow velocity from the Kolmogorov laws to yield:

$$S_v(\kappa) = 0.0365 C_k \varepsilon^{2/3} \kappa^{-5/3} \quad . \tag{6}$$

A characteristic length scale for turbulence is the integral length scale $L_v$. It is defined as

$$L_v = \frac{1}{\sigma_v^2} \int\limits_0^{\infty} dr B_v(r) \quad , \tag{7}$$

where $B_v(r)$ is the correlation function of flow velocity.

A model for atmospheric turbulence that extends to larger scales than the inertial subrange is the von Kármán model (von Kármán, 1948) which relates energy spectral density to the velocity variance $\sigma_v^2$ and the integral length scale $L_v$:

$$S_v(\kappa) = 2\sigma_v^2 L_v \left[ 1 + (8.42 L_v \kappa)^2 \right]^{-5/6} \tag{8}$$





## 3.2 Techniques to estimate turbulence dissipation rate

### 3.2.1 Sonic anemometers

TKE dissipation rate from the sonic anemometers on the meteorological towers $\varepsilon_s$ is calculated from the second-order structure function of the horizontal velocity (Eq. 5). $\varepsilon_s$ is calculated every $30\,s$, and the fit to the Kolmogorov model is done using a temporal separation between $\tau_1 = 0.1\,\text{s}$ and $\tau_2 = 2\,\text{s}$ (see also Bodini et al., 2018).

### 3.2.2 TLS

Estimates of $\varepsilon$ obtained by the TLS are retrieved using the inertial dissipation technique (Fairall et al., 1990). For each 1 s of data, a Hamming window was applied and the streamwise velocity spectra as a function of frequency was computed. The spectra was then smoothed and the mean structure function parameter $C_u^2$ was computed over the frequency band 5 to 10 Hz. The dissipation rate $\varepsilon_t$ was then computed using the Corssin relation:

$$\varepsilon_t = [0.52 C_u^2]^{3/2} \tag{9}$$

### 3.2.3 Profiling lidars

In case of a profiling lidar (or a scanning lidar used in a vertical stare mode as the OU CLAMPS lidar), TKE dissipation rate can be derived from the variance $\sigma_v^2$ of the line-of-sight (LOS) velocities (which in this case equals vertical velocity) following the approach described in (O'Connor et al., 2010) and further refined and validated in (Bodini et al., 2018). By assuming locally homogeneous and isotropic turbulence, the turbulence spectrum (Eq. 1) derived from the measured LOS velocity can be integrated within the inertial subrange:

$$\sigma_v^2 = \int\limits_{\kappa}^{\kappa_1} S_v(\kappa)d\kappa = -\frac{3}{2}\alpha\varepsilon^{2/3}\left(\kappa_1^{-2/3} - \kappa^{-2/3}\right) \tag{10}$$

For the OU CLAMPS lidar's vertical scans, which measured only the vertical component of velocity, the sample length $N$ to use for this integration is chosen by fitting the experimental spectra to the spectral model described in (Kristensen et al., 1989). Dissipation rate $\varepsilon_v$ can then be derived as:

$$\varepsilon_v = 2\pi\left(\frac{2}{3\alpha}\right)^{3/2}\left(\frac{\sigma_v^2 - \sigma_e^2}{L_N^{2/3} - L_1^{2/3}}\right)^{3/2} \tag{11}$$

where $L_1 = Ut$, with $U$ the horizontal wind speed and $t$ is the dwell time, $L_N = NL_1$ and $\alpha = 0.55$. The horizontal wind speed $U$ is retrieved from a sine-wave fitting from the velocity-azimuth display (VAD) scans, which were performed every 15 minutes. $\sigma_e^2$ accounts for the instrumental noise which affects the measured variance, and it is defined as in Pearson et al. (2009), using the technical parameters in Tab. 2. When the instrumental noise is too large, the inertial sub-range is difficult to detect in the lidar observations, and the dissipation retrievals are undermined. For this reason, for each spectral fit, we calculate





the deviation between the lidar spectrum $S_{\hat{v}}$ and the spectral model $S$ over the $n$ spectral frequencies used, and we quantify the error in the fit as:

$$E_S = \frac{1}{n}\sum_{i=1}^{n}\frac{|S_{\hat{v},i}-S_i|}{S_i} \tag{12}$$

Retrievals of $\varepsilon_v$ are discarded when $E_S > 10$. This threshold was chosen as it reliably removes noise-dominated spectra and provides the best agreement with the retrievals from other instruments as shown in Sect. 4.

### 3.2.4 RHI scans

Retrieving turbulence parameters from an RHI scan cannot be done with the same method as for vertical scans. Since the duration of a single scan is usually of the order of tens of seconds and in this experiment even one minute, it is not possible to
derive turbulence from the variance of LOS measurements only. The sampling time is too long to resolve the relevant scales in the inertial subrange in weak turbulence conditions. Here we propose an algorithm to retrieve eddy dissipation rate and integral length scales from RHI scans following the principal idea of Smalikho et al. (2005). We introduce a modification of the Doppler Spectrum Width (DSW) method which uses variance of LOS velocities $\sigma_v^2$ and the turbulent broadening of the Doppler spectrum $\sigma_t^2$. In Smalikho et al. (2005), the RHI scans are used to calculate vertical profiles of $\varepsilon$ by binning data points
from the RHI scan into height bins. The complex flow over the Perdigão double ridges compromises this approach. To observe variability of turbulence over the complex terrain, we divide the area covered by the RHI scan into square sub-areas with a defined side length (here: $s_a = 20\,m$). Within these sub-areas, the LOS variance is calculated as a space and time average over a half-hour period:

$$\hat{\sigma}_v^2 = \frac{1}{N}\sum_{i=0}^{N}[\hat{v}_{r,i}-\overline{v}_r]^2 \quad , \tag{13}$$

where $N$ is the number of single LOS measurements within the time and space bin and $\overline{v}_r$ is the mean of all measurements in the bin. Variables with hat denote measured variables. The half-hour averaging period has been chosen as a common averaging time for turbulence measurements in the ABL. Longer periods could be affected by the mesoscale changes of the flow field and shorter periods reduce the number of single RHI scans, which increases the uncertainty of variance measurements.
The turbulent broadening of the Doppler spectrum is defined as:

$$\sigma_t^2 = \hat{\sigma}_{sw}^2 - \sigma_0^2 - \hat{\sigma}_s^2 - E \tag{14}$$

with $\hat{\sigma}_{sw}^2$ the measured spectral width, $\sigma_0^2$ the spectral width at constant wind speed in the sensing volume, $\hat{\sigma}_s^2$ the measured spectral broadening caused by shear, and $E$ the random error. According to Smalikho et al. (2005), we set the noise threshold for derivation of parameters from the Doppler spectrum to $n_{th} = 1.01$. This value is much smaller than for the lidar used in Smalikho et al. (2005) due to the high number of accumulations by the fiber-based system used in this study. We then assume
$E$ to be negligible.
To maximize the signal-to-noise ratio of the spectra and thus estimate more reliable spectral widths at low signal strength,





all the spectra within a time and space bin are interpolated in the Fourier domain, aligned according to their maxima, and accumulated. The spectral width of the accumulated spectra is used as $\hat{\sigma}_{\mathrm{sw}}^2$.

The contribution from shear $\hat{\sigma}_s^2$ is calculated according to Smalikho et al. (2005) for each LOS measurement and averaged over
the time and space bin:

$$\hat{\sigma}_s^2 = \frac{1}{2\pi} \left[ \frac{\frac{1}{N}\sum_{i=0}^{N}(\hat{v}_{r,i}(r+\Delta R)-\hat{v}_{r,i}(r-\Delta R))\Delta z}{2\Delta R} \right]^2 \quad , \tag{15}$$

where $N$ is the number of LOS measurements in the time and space bin, $\hat{v}_{r,i}(r)$ is the measured radial velocity of the range gate at location $r$, $\Delta R$ is the distance between two range gates, and $\Delta z$ is the physical resolution of the lidar measurement. The spectral width at zero wind speed $\sigma_0^2$ as well as $\Delta z$ can be theoretically derived from the lidar parameters $T_w$ (time window) and $\sigma_p$ (pulse width) through a model of the Doppler lidar echo signal as described in Smalikho et al. (2013). The
echo signal models assume a specific pulse shape and require knowledge of the lidar parameters $T_w$ and $\sigma_p$. These vary for different systems and are only given as estimations by the lidar manufacturer. Here, we will consider $\sigma_0$ and $\Delta z$ as unknown parameters that need to be tuned within physically reasonable limits to achieve good agreement with reference instruments. As an initial guess for $\sigma_0$, the mean of all observed spectral widths can be used. For $\Delta z$, the initial guess is the physical resolution as provided by the manufacturer for the used lidar settings (in this case, 50 m). It has to be noted that since the calibration of
these parameters will also account for inaccuracies in the assumptions made for the theoretical turbulence model, the estimated parameters are not necessarily the real physical lidar parameters.

A model for the volume averaging of the lidar measurement and basic turbulence theory as described above is used to derive the equations for the retrieval of turbulence from RHI scans. Assuming that the lidar pulses have Gaussian shapes, a window function for LOS measurements of wind speed can be defined as

$$Q_s(z) = \frac{1}{\Delta z}e^{-\pi z^2/\Delta z^2} \quad , \tag{16}$$

so that the wind speed measured by the lidar is the convolution of the actual wind speed with the window function

$$\hat{v}(r) = \int\limits_{-\infty}^{\infty} dz\, Q_s(z) v(r+z) \quad . \tag{17}$$

The transfer function of the low-pass spatial filter of the lidar, derived from the window function, is

$$H_p(\kappa) = \left[ \int\limits_{-\infty}^{\infty} dz\, Q_s(z) e^{-2\pi j\kappa z} \right]^2 \quad . \tag{18}$$

The total variance of the LOS velocity $\sigma_v^2$ is the sum of measured variance $\hat{\sigma}_v^2$ and turbulent broadening of the spectra $\sigma_t^2$:

$$\sigma_v^2 = \hat{\sigma}_v^2 + \sigma_t^2 \quad . \tag{19}$$




The variances are the integral of the power spectra multiplied by the respective filter function:

$$\hat{\sigma}_v^2 = 2 \int_0^\infty d\kappa\, S_v(\kappa) H_p(\kappa) \quad , \tag{20}$$

$$\sigma_t^2 = 2 \int_0^\infty d\kappa\, S_v(\kappa) \left[1 - H_p(\kappa)\right] \quad . \tag{21}$$

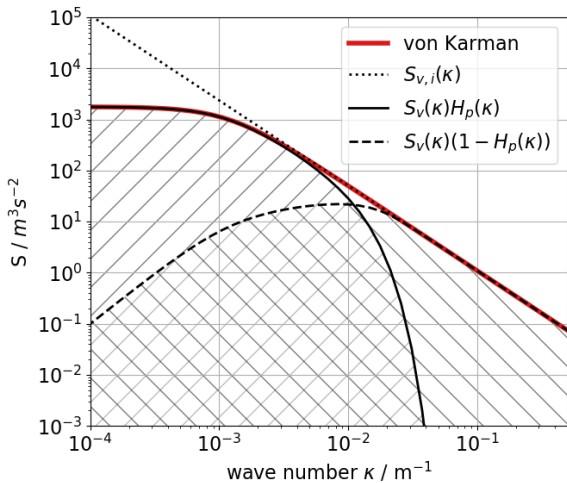

**Figure 3.** Theoretical spectrum for atmospheric turbulence and the contributing filtered spectra as measured by a lidar. The hatched areas show the areas for the integration to calculate $\hat{\sigma}_v^2$ ('/') and $\sigma_t^2$ ('\') respectively, whereas the integration of the area under the red curve yields $\sigma_v^2$.

A generic spectrum, subdivided into areas of lidar-measured variances, appears in Figure 3.

5 Substituting $S_v$ for the von Kármán model, we obtain

$$\sigma_t^2 = 4\sigma_v^2 L_v \int_0^\infty d\kappa \frac{\left[1 - H_p(\kappa)\right]}{\left[1 + (8.42 L_v \kappa)^2\right]^{5/6}} \quad . \tag{22}$$

In the inertial subrange of turbulence, the van Kármán model can be simplified to

$$S_{v,i}(\kappa) = 2\sigma_v^2 L_v (8.42 L_v \kappa)^{-5/3} \quad , \tag{23}$$

and in combination with Eq. 6, it can be solved for $\sigma_v^2$:

$$\sigma_v^2 = \frac{C_k}{1.573} \varepsilon^{2/3} L_v^{2/3} \quad . \tag{24}$$

Substituting $\sigma_v^2$ in Eq. 22 with Eq. 24 is

$$\sigma_t^2 = 2.54 C_k \varepsilon^{2/3} L_v^{5/3} \int_0^\infty d\kappa \frac{\left[1 - H_p(\kappa)\right]}{\left[1 + (8.42 L_v \kappa)^2\right]^{5/6}} \quad . \tag{25}$$





With the simplified equation for $H_p$

$$H_p(\kappa) = \exp\left(-2\pi(\Delta z \kappa)^2\right) \quad , \tag{26}$$

and substitution of $\kappa$ for $\xi = 2\pi \Delta z \kappa$, the equation can be rewritten as a function of $\Delta z$ and $L_v$:

$$\sigma_t^2 = 0.2485 C_k \varepsilon^{2/3} \Delta z^{2/3} \int\limits_0^\infty \frac{d\xi \left(1 - \exp\left[-\xi^2/(2\pi)\right]\right)}{\left[\xi^2 + (0.746\Delta z/L_v)^2\right]^{5/6}} \tag{27}$$

Substituting $\varepsilon$ in Eq. 27 with the solution for $\varepsilon$ from Eq. 24, the equation for $\sigma_t^2/\sigma_v^2$ as a function of $\Delta z$ and integral length scale $L_v$ as they appear in Smalikho et al. (2005) can be formulated:

$$\sigma_t^2/\sigma_v^2 = F_w(L_v, \Delta z) \tag{28}$$

$$F_w(L_v, \Delta z) = (1.972)^{2/3} C_k^{-1} L_v^{-2/3} G_w(\Delta_z, L_v) \tag{29}$$

$$G_w(\Delta_z, L_v) = 0.2485 C_k \Delta z^{2/3} \int\limits_0^\infty \frac{d\xi \left(1 - \exp\left[-\xi^2/(2\pi)\right]\right)}{\left[\xi^2 + (0.746\Delta z/L_v)^2\right]^{5/6}} \tag{30}$$

The only unknown is $L_v$. A downhill-simplex algorithm is used to minimize Eq. 28 for $L_v$

$$\underset{L_v}{\arg\min}\left[-exp\left(-(F_w(L_v, \Delta z) - \frac{\sigma_t^2}{\sigma_v^2})^2\right)\right] \quad . \tag{31}$$

Minimization of Eq. 31 is computationally expensive. To accelerate the data processing, a power function can be defined which approximates the relationship between $L_v$ and $\sigma_t^2 \sigma_v^{-2}$:

$$\hat{L}_v = c_1 \left(\frac{\sigma_t^2}{\sigma_v^2}\right)^{c_2} + c_3 \quad . \tag{32}$$

The coefficients $c_1$, $c_2$ and $c_3$ are determined by a curve fit over the range of $L_v = 3..1000$m to Eq. 28. The coefficients are specific for each lidar, since they depend on $\Delta z$, and will be determined in Sect. 4. The fitting curve and residuals as obtained through minimization of Eq. 31 appear in Fig. 4. It shows that the error $L_v - \hat{L}_v$ that is made with the power-law approximation is in the range of $\pm 1$ m. For simplicity, we will use the variable name $L_v$ for integral length scales calculated with Eq. 32 in the following.

With $L_v$ and measured variance $\sigma_v^2$ in Eq. 24, eddy dissipation rate for the RHI measurements $\varepsilon_r$ can finally be calculated as

$$\varepsilon_r = \frac{1.972}{C_k^{3/2}} \frac{\sigma_v^3}{L_v} \quad . \tag{33}$$

For optimal accuracy of the dissipation rate retrieval, the two unknowns $\sigma_0$ and $\Delta z$ need to be calibrated according to reference instruments. In this study, the sonic anemometer at 100 m on tower 25/trSE09 is the closest in-situ observation to the RHI scans in the valley. It is used for the calibration by minimizing the root mean square error between this measurement and the respective RHI scan. The resulting parameters differ slightly for lidars DLR#1 and DLR#2 and appear in Tab. 3 along with the coefficients for the power-law fit of $L_v$. The difference in $\Delta z$ and $\sigma_0$ can be partially attributed to instrumental variability but will also incorporate other sources of error in the turbulence model and data retrieval.




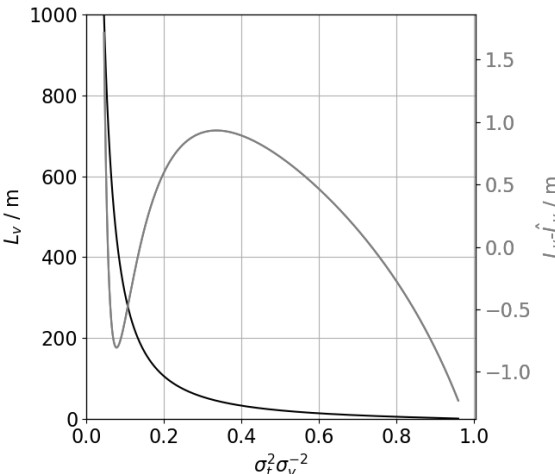

**Figure 4.** Dependency of $L_v$ on $\sigma_t^2 \sigma_v^{-2}$ according to Eq. 28 (black curve) and the residuals of the power-law fit according to Eq. 32 (gray line).

**Table 3.** Adjusted parameters for dissipation rate retrieval from RHI scans.

|  | $\sigma_0^2$ | $\Delta z$ | $c_1$ | $c_2$ | $c_3$ |
|---|---|---|---|---|---|
| DLR#1 | 1.218 | 20.689 | 6.072 | -1.4892 | -4.873 |
| DLR#2 | 1.621 | 23.272 | 6.858 | -1.4879 | -5.584 |

## 3.3 Estimation of uncertainties

### 3.3.1 Sonic anemometers and TLS

15 To estimate the uncertainty of the retrievals of $\varepsilon_s$, we apply the law of combination of errors, which describes how random errors propagate through a series of calculations (Barlow, 1989). For a function $g = g(x_i)$, with $x_i$ the independent and uncorrelated variables, the law of combination of errors states that, for small errors (i.e. if we ignore second order and higher terms), the variance of the function $g$, approximated by the sample variance $\sigma_g^2$, is given by:

$$\sigma_g^2 = \left(\frac{\partial g}{\partial x_i}\right)^2 \sigma_{x_i}^2 \tag{34}$$

20 where $\sigma_{x_i}^2$ are the sample variances of the $x_i$. By applying this method to equation (5), the fractional standard deviation in the $\varepsilon$ estimate is (Piper, 2001):

$$\sigma_{\varepsilon,s} = \frac{3}{2}\frac{\sigma_I}{I}\varepsilon \tag{35}$$





where $I$ is the sample mean of $\tau^{-2/3}D(\tau)$, and $\sigma_I^2$ is its sample variance.

Similarly, uncertainties are calculated for the TLS measurements $\varepsilon_t$. However, since the determination of dissipation rate is done in the frequency domain, $I$ in this case is the sample mean of $\kappa^{5/3}S(\kappa)$.

### 3.3.2 Profiling lidar

The uncertainty in the $\varepsilon_v$ retrievals from the profiling lidars can be estimated from the uncertainty of the LOS velocity variance
by also applying the law of combination of errors (Eq. 34) to Eq. 11:

$$\sigma_{\varepsilon,v} = \left| \frac{\partial \varepsilon_v}{\partial \sigma_v} \right| \sigma_{\sigma,v} = 2\pi \left( \frac{2}{3\alpha} \right)^{3/2} \left( \frac{\sigma_v^2 - \sigma_e^2}{L_N^{2/3} - L_1^{2/3}} \right)^{1/2} \frac{3\sigma_v}{L_N^{2/3} - L_1^{2/3}} \sigma_{\sigma,v} \tag{36}$$

$$= \varepsilon_v \frac{3\sigma_v}{\sigma_v^2 - \sigma_e^2} \sigma_{\sigma,v} \tag{37}$$

where $\sigma_{\sigma,v}$ is the uncertainty of the sample variance. This value is not known, but is considered to be of the same order of magnitude as the instrument noise and thus set to the value of $\sigma_e$.

### 3.3.3 RHI

The uncertainty of $\varepsilon_r$ can be calculated by Gaussian uncertainty propagation through Eq. 33 if the uncertainties of the estimation of the integral length scale $\sigma_{L,v}^2$ and the uncertainty of the measurement of the wind speed variance $\sigma_{\sigma,v}$ are known:

$$\sigma_{\varepsilon,r} = \sqrt{ \left( \frac{5.916\sigma_v^2}{C_k^{3/2} L_v} \sigma_{\sigma,v} \right)^2 + \left( -\frac{1.972\sigma_v^3}{C_k^{3/2} L_v^2} \sigma_{L,v} \right)^2 } \quad . \tag{38}$$

The uncertainty of the variance of radial velocities $\sigma_{\sigma,v}$ can be determined from the uncertainty of the turbulent broadening estimation $\sigma_{\sigma,t}$ and the uncertainty of measured LOS velocities $\sigma_{\hat{\sigma},v}$:

$$\sigma_{\sigma,v} = \sqrt{\sigma_{\sigma,t}^2 + \sigma_{\hat{\sigma},v}^2} \tag{39}$$

The uncertainty of the measurement of the integral length scale $L_v$ cannot be determined directly from Eq. 31. A propagation of uncertainties is not possible here, because the function is not differentiable. The approximated function Eq. 32 can, however,
be differentiated with respect to $\sigma_v$ and $\sigma_t$ and can thus be used to propagate uncertainties of the measured values to $L_v$, so that

$$\sigma_{L,v} = \sqrt{ \left[ 2c_1 c_2 \left( \frac{\sigma_t}{\sigma_v^2} \right)^{c_2} \sigma_{\sigma,v} \right]^2 + \left[ -2c_1 c_2 \left( \frac{\sigma_t^2}{\sigma_v^3} \right)^{c_2} \sigma_{\sigma,t} \right]^2 } \tag{40}$$

with $c_1, c_2$ and $c_3$ from Tab. 3. The uncertainties that are found through this approach are then fed into Eq. 38 in order to calculate the uncertainty of $\varepsilon_r$.

As can be seen from Eq. 38, the uncertainty of the retrieval of $\varepsilon_r$ depends strongly on the combination of $\sigma_v$ and $\sigma_t$. A two-dimensional map visualizing the relative error $\sigma_{\varepsilon,r} \varepsilon_r^{-1} 100\%$ appears in Figure 5. The contour lines show that uncertainties



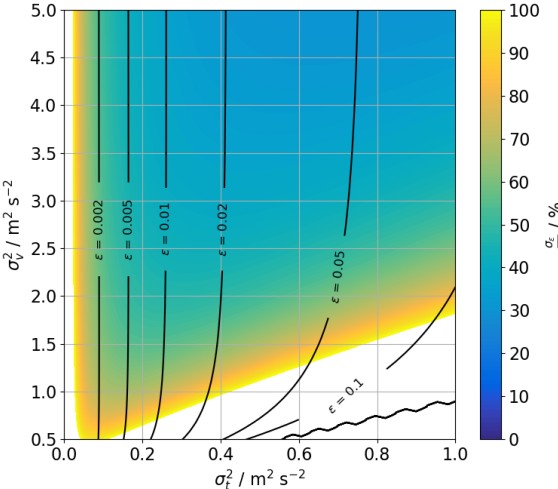

**Figure 5.** Uncertainty estimation for dissipation rate $\varepsilon_r$ in dependency of measured variances $\sigma_v^2$ and $\sigma_t^2$. The contour lines show the associated $\varepsilon_r$.

grow very large for dissipation rates smaller than $10^{-3}$ m$^2$s$^{-3}$. Uncertainties are also large for values in excess of $10^{-1}$ m$^2$s$^{-3}$ if $\sigma_v^2$ is small at the same time. The input uncertainty $\sigma_{\sigma,v}$ depends on the CNR and is assumed to be of the order of the corresponding instrumental noise $\sigma_e$. The values in Fig. 5 are calculated with input uncertainties $\sigma_{\sigma,v} = \sigma_{\sigma,t} = 0.05$ m s$^{-1}$ which correspond to a CNR-value of approximately -11 dB, which is common for the signal strength during the Perdigão campaign for the DLR lidars.

## 4 Validation and intercomparison

Here we demonstrate a single-point comparison between in-situ and remote sensing retrievals of TKE dissipation rate, and we
5   then compare the different methods via vertical profiles of $\varepsilon$.

### 4.1 Single-point validation

Sonic anemometer measurements of eddy dissipation rate are continuously available throughout the campaign. Even though the location of tower 25/trSE09 is approximately 150 m up-valley from the RHI plane and ~250 m up-valley from the Lower Orange Site, where the vertical stare and TLS measurements are taken, we expect a similar development of turbulence at 100 m
10   above ground on average from all the instruments considering that they are all within the center of the valley.

Vertical stare data from OU CLAMPS are available from 6 May through 15 June 2017. Half-hour averaged dissipation rate retrievals are compared with the measurements from the sonic anemometer in the scatter plot in Fig. 6(a). To compare data at the same height, the lidar results have been interpolated to 100 m above ground. The sonic anemometer and CLAMPS vertical



stare measurements correlate with a coefficient of $R = 0.58$, but have significant scatter, which can likely be attributed to the

15 spatial separation between the two instruments and the heterogeneity of complex terrain flow. The scatter grows in the low-turbulence regime, where the uncertainty of the retrievals increases.

DLR lidars #1 and #2 operated with the same parameters from 9 June to 15 June 2017. Figure 6(b) compares their RHI estimates with the sonic anemometer estimates for all the valid half-hour measurements in this period. Since the sonic anemometer has been used for calibration of the RHI retrieval, no biases between the measurements can occur. The correlation between the measurements resembles that found for the vertical stare measurements ($R = 0.62$), with a larger scatter in low turbulence cases.

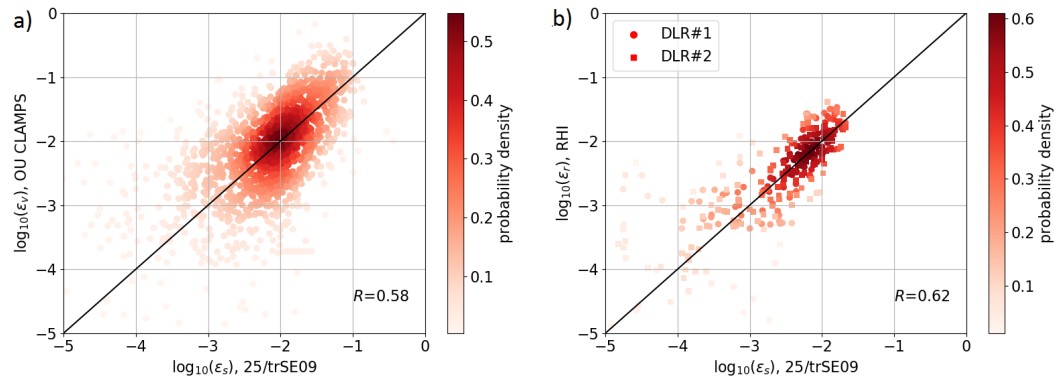

**Figure 6.** Comparison of half-hour averaged estimates of $\varepsilon$ at 100 m above ground between the sonic anemometer on tower 25/trSE09 and the OU CLAMPS vertical stare lidar measurements (a) as well as for the same sonic anemometer and DLR#1 and DLR#2 (b). The color scales represent the density of probability of a measurement point. The black line is the line of identity.

Between 13 June, 2100 UTC and 14 June, 1200 UTC data are available from all the instruments, and the TLS made multiple successive ascents and descents. During this night, a low-level jet (LLJ) from the south-west occurred, with its peak wind speed

5 at varying heights between 200 m and 400 m above ridge height, inducing shear and veer within and above the valley. The two-dimensional wind field over the valley, as reconstructed from the RHI-scans of lidars DLR#1 and DLR#2 (Wildmann et al., 2018a), for an averaging period of 30 minutes, appears in Fig. 7. App. B presents a detailed description of the atmospheric conditions for this LLJ.

The eddy dissipation rate generally increases over this time period (Figure 8), with consensus between the instruments except for an interesting period between 0430 UTC and 0700 UTC. For those systems that resolve parts of the inertial sub-range of turbulence, the variance spectra appear in Fig. C1 in App. C. Dissipation rate from the TLS is calculated for a time series

5 corresponding to a height bin between 90 m and 110 m above ground during its ascents and descents to facilitate comparison with the 100-m tower measurements. The different instruments within the valley generally concur, while the tower on the ridge suggests very different (and smaller) values of dissipation. The best agreement emerges between the CU TLS and OU CLAMPS, which both measure approximately at the same location. In some periods OU CLAMPS estimates deviate from





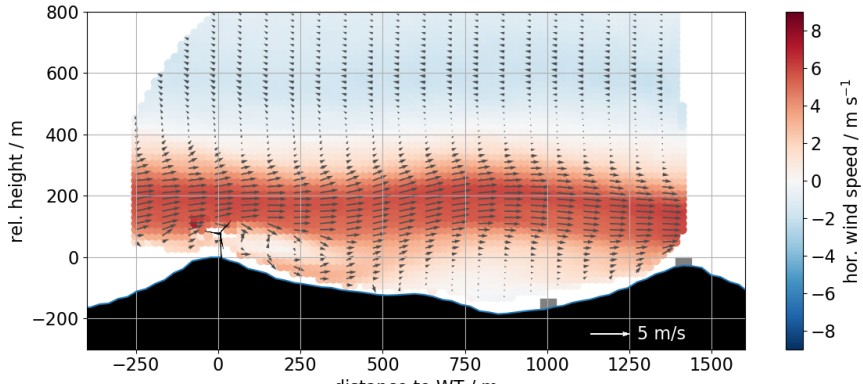

**Figure 7.** Wind field visualization for 14 June 2017, 0130-0200 UTC. The arrows show the wind vectors projected onto the RHI plane as retrieved with the coplanar method (Wildmann et al., 2018a). The colormap is scaled with the horizontal wind component of the projected wind vector. The origin of the local coordinate system in this and all the following plots is relative to the wind turbine base location on the SW ridge.

the other systems, which can potentially be attributed to specific wind directions and local turbulence features. One period in which all systems show strong variability and divergent measurements is between 0430 UTC and 0700 UTC which is also the time when TLS measurements are available. This period will be analyzed in detail in Sect. 5. The comparison between the two towers on the ridge and in the valley shows that during the night, turbulence in the valley is decoupled from the flow above ridge height, while at sunrise, just after 0700 UTC, the measured values of $\varepsilon$ converge. Similarly, wind speed values (Figure 8c) between the ridge and valley also differ through the night until convergence after sunrise.

## 4.2 Comparison of vertical profiles in the valley

To investigate this decoupling more closely, we assess profiles of TKE dissipation rate. Both TLS and lidars allow for collection of estimates throughout the whole ABL. Therefore, the TLS and the lidars enable the estimates of turbulence from the network of sonic anemometers to extend to higher altitudes. Vertical profiles of TKE dissipation rate can then be measured both in and over the valley, which is particularly important when assessing turbulence in nighttime LLJ flows. The average vertical profiles of $\varepsilon$ as measured by the OU CLAMPS vertical stares, the DLR RHIs and the CU TLS for two selected time periods (0400-0430 UTC and 0500-0530 UTC) appear in Fig. 9. The sonic anemometer measurements at all levels on towers 20/trSE04 and 25/trSE09 are also included in the profiles. Data from the lidars and the sonic measurements represent half-hour averages, whereas the TLS measurements are quasi-instantaneous, with a moving spatial filter of approximately 20 m for the ascents and descents at constant speed. Since the collection of a full profile at an ascent of 0.3 ms$^{-1}$ lasts approximately 23 minutes, these high-resolution measurements suggest some idea about the variability of $\varepsilon$ within the averaging period of the other systems. The gap in the measurements between 200 m and 400 m above the ridge-top height are due to low turbulence in that region which



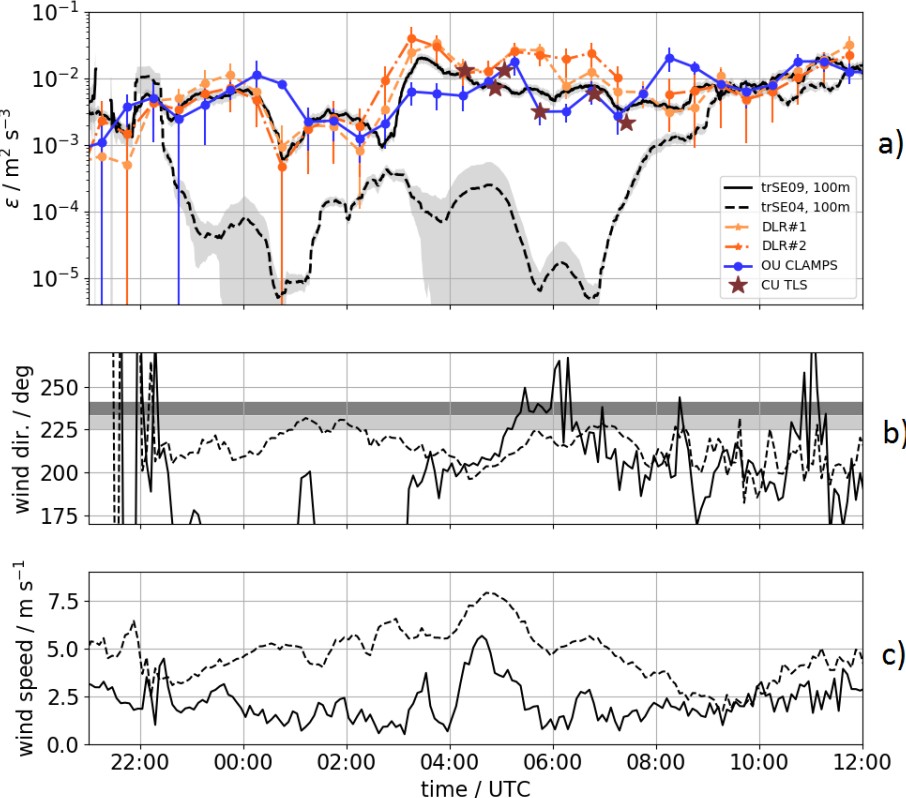

**Figure 8.** Comparison of $\varepsilon$ as measured by the RHI scans, the vertical stare measurements, the TLS and two reference sonic anemometers on 14 June 2017, 0000-1200 UTC (a). Panel (b) shows the wind direction and (c) the wind speed measured by sonic anemometers at the 100 m level over the ridge and in the valley. The shaded areas in (b) give the wind direction regions in which OU CLAMPS (light grey) and DLR#1 and #2 (dark grey) are in the line-of sight of the WT rotor plane.

could not be adequately sampled with the lidars and thus filtered according to the criteria defined in Sect. 3. The upper limit of TLS measurements was limited by flight permissions. All instruments indicate a large gradient of turbulence at ridge height, with values of $\varepsilon$ at 100 m above the ridge almost two orders of magnitude smaller than in the valley. At 0400..0430 UTC, a large variability among the different platforms occurs, when the LLJ is still well-defined, with maximum wind speed at 300 m above ridge height. One hour later, with a LLJ that is broadening and weakening, the vertical profiles of all systems agree better

5 above the ridge. Moreover, as also seen in Fig 8, all valley instruments measure increased turbulence in the valley except for the sonic anemometers, as will be explored in more detail in Sect. 5.

To quantify systematically the agreement between RHI and vertical stare retrievals of dissipation rate, all valid measurement points between -150 m and 800 m above ridge height at the location where the vertical stares are taken can be compared, for 14





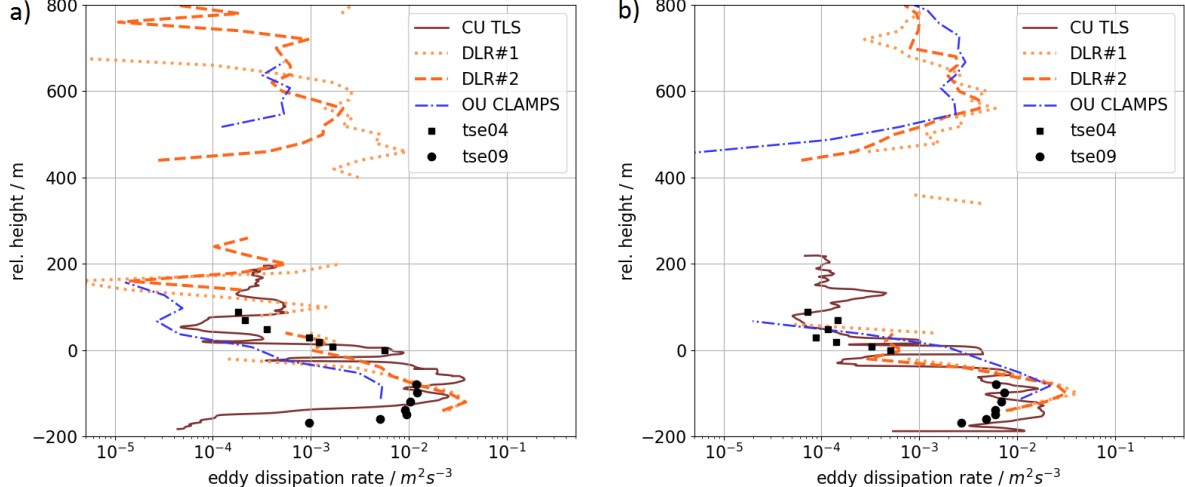

**Figure 9.** Comparison of average vertical profiles of eddy dissipation rate $\varepsilon$ measured by lidar vertical stare, RHI scans, TLS and sonic anemometers on meteorological towers at 0400-0430 UTC (a) and at 0500-0530 UTC (b).

June 2017. For this purpose, the values of both systems are interpolated to the same heights and compared (Fig. 10). As in the comparison with the sonic anemometer, the correlation is high (R=0.55), with a larger scatter for values of $\varepsilon$ less than $10^{-3}$.

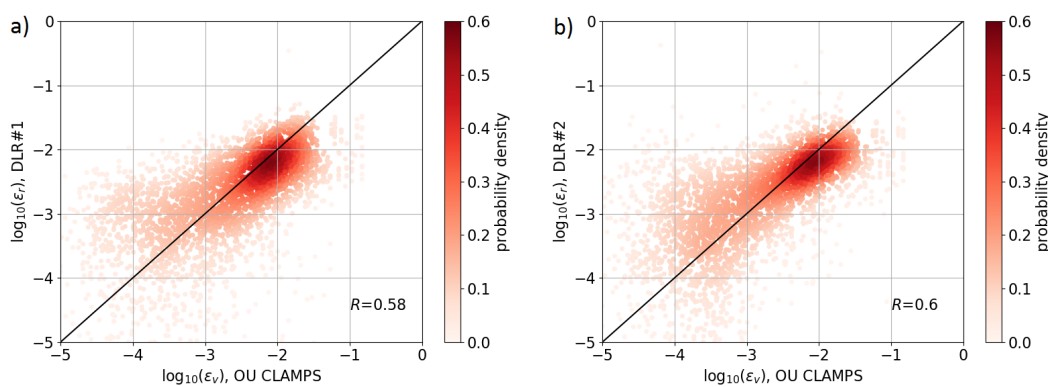

**Figure 10.** Comparison of all estimates of $\varepsilon$ in the vertical profile over the valley from 09-15 June 2017 from DLR#1 (a) and DLR#2 (b) against the Halo system. The color scale represents the probability of occurrence of a measurement point. The black line shows identity.

Estimates of $\varepsilon_r$ should not depend on the direction of the lidar beams. These directions differ significantly for the two lidars. A comparison between retrievals of $\varepsilon_r$ from the two lidars performing RHI scans in the whole observed area for the period 13

5    June, 2000 UTC to 14 June, 1200 UTC (Fig. 11) shows a very good agreement between $10^{-3}$ $m^2 s^{-3}$ and $10^{-1}$ $m^2 s^{-3}$. Again, a rather large scatter occurs in the region of low turbulence, consistent with all the instruments.





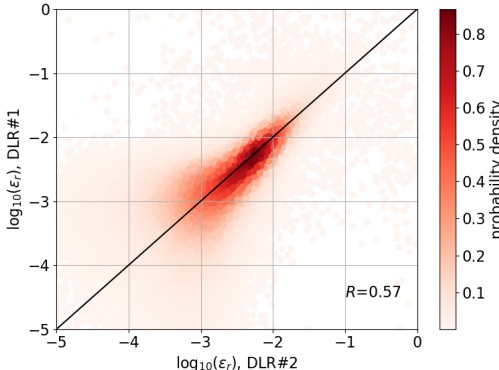

**Figure 11.** Comparison of all estimations of $\varepsilon$ in the RHI plane for 14 June 2017 between DLR#1 and DLR#2. The color scales represent the probability of occurrence of a measurement point. The black line is the line of identity.

The agreement between the different systems can also be confirmed by plotting the vertical profiles of the vertical stare $\varepsilon_v$ and TLS measurements $\varepsilon_t$ on top of the calculations of $\varepsilon_r$ from DLR#1 and DLR#2 (Fig. 12) for the periods of main interest for our study. The larger turbulence in the shear layers at the upper and lower bound of the LLJ emerge clearly in this representation. Missing data points occur in the very low turbulence regions in the center of the jet and above 600 m above the ridge top in Fig. 12. At these points, the Doppler spectral width becomes too small to be distinguished from noise, i.e. the value of $E$ in Eq. 14 is not negligible any more.

## 5   Case study - wind turbine wake

Despite the general good agreement between all instruments, some time periods show significant differences, such as between 0430 UTC and 0700 UTC on 14 June, seen in the time series at 100 m (Fig. 8). Here we present evidence that this disagreement arises because of spatial heterogeneity in turbulence related to the propagation of the wind turbine wake within the measurement domain.

### 5.1   Wind turbine wake turbulence

During this time period, wind speeds at the SW ridge (tower 20/trSE04, Fig. 8(c)) exceeded $5\,\mathrm{m\,s^{-1}}$ which is well within the power-production range of the WT and generation of a wind turbine wake can be expected. From Fig. 8(b) we can see that the local wind direction steers the wake toward the measurement volumes of the instruments discussed here: into the region measured by the OU CLAMPS and TLS near 0515 UTC and into the DLR RHI plane after 0545 UTC.

At 0515 UTC, vertical profiles of wind speed, wind direction and potential temperature as observed by multiple instruments provide insight into the steering of the wake (Fig. 14). In addition to the TLS, a radiosonde, the tower 25/trSE09 and the OU CLAMPS VAD measurements, which are all located in the valley center, Fig. 14 also includes data from the tower 20/trSE04



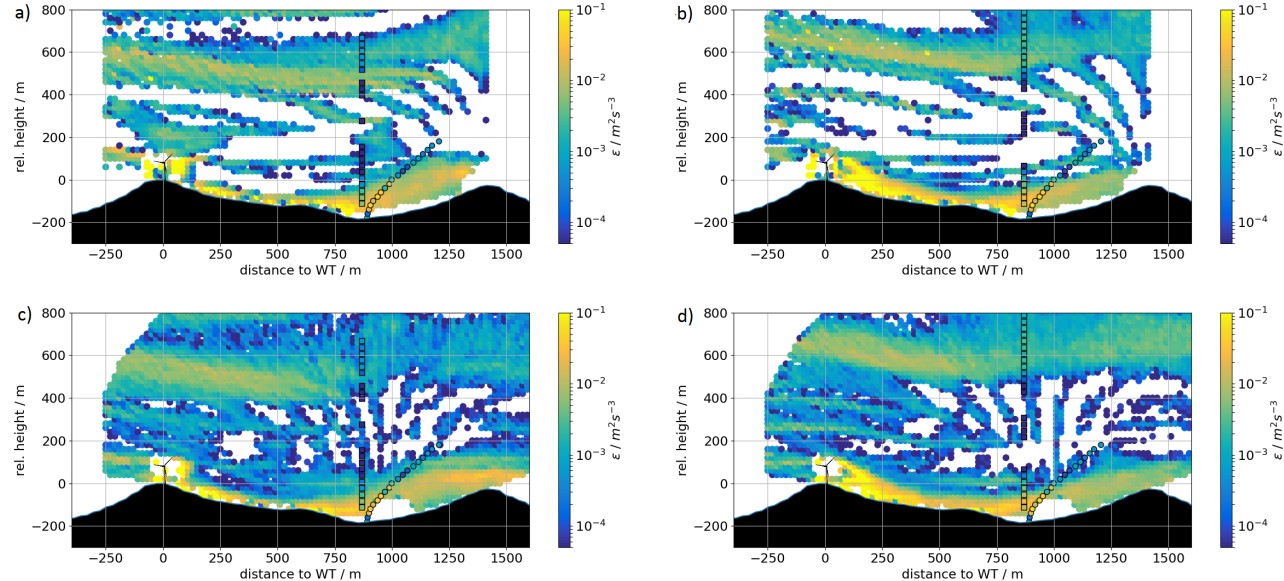

**Figure 12.** Measurements of vertical stare and TLS on RHI scans by DLR#1 (a and b) and DLR#2 (c and d) for the time period 0400-0430 UTC (a and c) and 0500-0530 UTC (b and d).

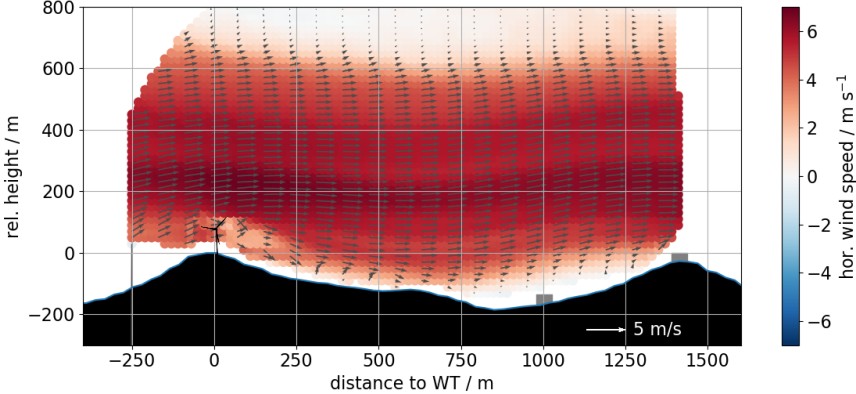

**Figure 13.** Coplanar wind field reconstruction from DLR#1 and DLR#2 from averaged RHI scans between 0530 UTC and 0600 UTC.

on the SW ridge next to the turbine, as well as half-hour averages of virtual towers (VT) calculated at the intersection lines of the RHIs of all three DLR lidars, including DLR#3 (for details about the VT method, see Bell et al. (2019)). The VTs provide a wind estimate downwind of the wind turbine, at four distinct locations, each separated by one rotor diameter $D$ (as highlighted in Fig. 1). The vertical profiles of wind direction from the four VTs match each other down to a height of 100 m above ridge height. Below this, winds veer with different strength, depending on the location on the sloping valley transect. At the 100 m



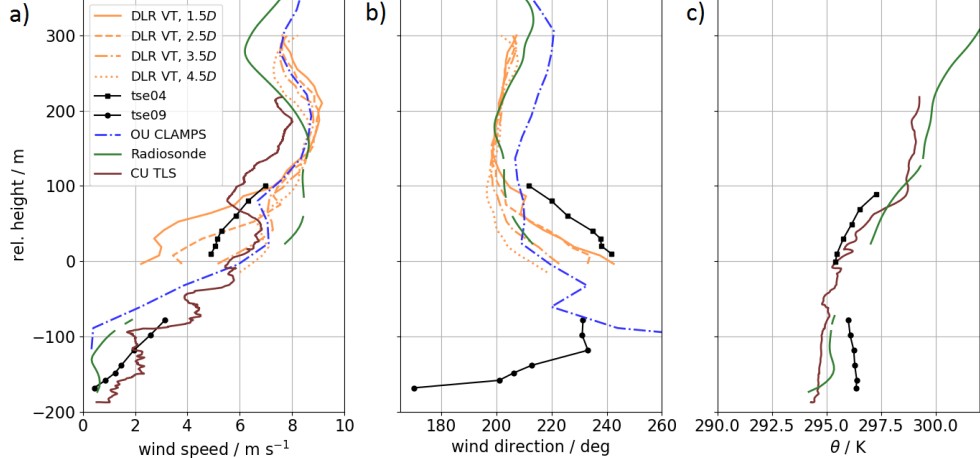

**Figure 14.** Vertical profiles of wind speed (a), wind direction (b) and potential temperature $\theta$ (c) from 0500-0530 UTC. For wind speed, tower data of towers 20/trSE04 and 25/trSE09 is complemented with profiles of the DLR VTs. For $\theta$, TLS measurements are included.

level of tower 25/trSE09 (within the valley), a wind direction of $225°$ is measured aligning the wind turbine with the Lower Orange Site. Potential temperature measurements by the radiosonde that was released at 0516 UTC in the valley and by the CU TLS clearly show a remaining nighttime inversion capping the boundary-layer flow approximately 100 m above ridge height.

10 At 0545 UTC, turbulence retrievals from the DLR RHI measurements (Fig. 8) suggest large turbulence levels in the Vale do Cobrão, whereas the rest of the instruments observe significantly lower turbulence. This corresponds to a wind direction in the valley that has veered further towards the RHI plane in a wind direction of $235°$. At the same time, the coplanar wind retrievals in the RHI plane show a wind turbine wake with clearly detectable wind speed deficit (Fig. 13) in the first 250 m near the turbine. Further downstream, wind speed deficits of the wake are hard to distinguish from the ambient flow.

15 To better understand the three-dimensional propagation of the WT wake into the valley, measurements of lidar DLR#3 give information about the location of the WT wake perpendicular to the RHI plane. Figure 15 shows the measured mean turbulent broadening $\sigma_t^2$ from the DLR#3 lidar for the scans at a distance of 1.5 $D$ (a) and 4.5 $D$ (b) from the WT. No dissipation rates were calculated from these measurements, because less than ten RHI scans for each azimuth angle could be performed within 30 minutes, thus not providing robust enough statistics for the calculation of the LOS variance $\hat{\sigma}_v^2$. However, a ring of large $\sigma_t^2$ can clearly be noticed at the expected wake location at a distance of 1.5 $D$ from the WT, which is still in the near-field of the WT. At the more distant RHI scan (4.5 $D$), an ellipsoid region of higher $\sigma_t^2$ occurs. While the top of the wake is found at a viewing angle of $210°$, the bottom of the wake stretches out from $215°$ to $235°$, thus suggesting that the wake is advected and stretched with the veering mean wind as it propagates down the valley. Numerous previous observations (Bodini et al., 2017)

and simulations (Vollmer et al., 2017) indicate that wakes veer in response to ambient veer.

The 2D-plots of $\varepsilon_r$ by DLR#1 and #2 (Fig. 16) support the theory of wake-induced turbulence being trapped under the inversion and propagating into the valley by showing constantly large turbulence between WT and the valley and even some indication of





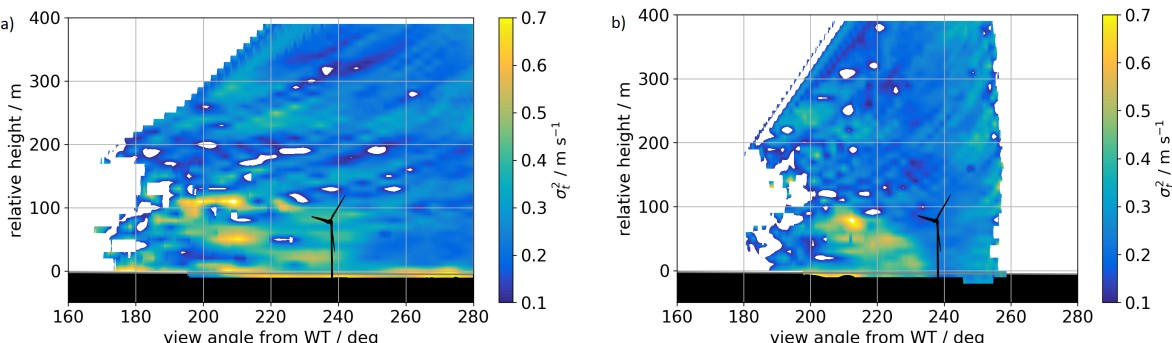

**Figure 15.** Average turbulent broadening of the measured Doppler spectra from DLR#3 for 0530-0600 UTC. The left panel shows the results of the RHI cutting the wake at 1.5 $D$, the right panel at 4.5 $D$. The x-axis corresponds to the viewing angle centered at the WT location.

the expected tip vortex turbulence in the WT near-field. The dissipation rate measured in the waked region by the lidar systems

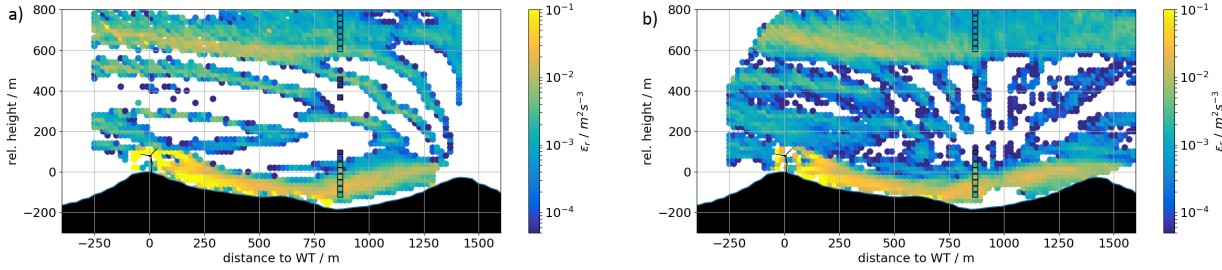

**Figure 16.** RHI and vertical stare measurements of eddy dissipation rate for 0545 UTC showing strong evidence of wind turbine wake-induced turbulence in the valley.

more than 10 rotor diameters downstream of the WT is approximately $2 \cdot 10^{-2}$ m$^2$s$^{-3}$, compared to $6 \cdot 10^{-3}$ m$^2$s$^{-3}$ measured
by the sonic anemometer in the region less effected by the wake-induced turbulence. These differences are smaller than the differences of two orders of magnitude within-wake and out-of-wake by Lundquist and Bariteau (2015), but the Perdigão measurements are much further downwind than the Lundquist and Bariteau (2015) measurements.

# 6   Assessment of the results

## 6.1   Turbulence measurement in complex terrain

This study demonstrates how measurements of multiple instruments can be synthesized to evaluate the spatio-temporal evolution of turbulence in a highly complex terrain. Vertical profiles retrieved from vertical stare measurements of a scanning lidar compare well with in-situ measurements from meteorological masts and a TLS, extending the upper limit of the vertical profile




significantly. The new retrieval method for TKE dissipation rate from RHI scans allows localization of the origin of turbulence in a way that would not be possible with point measurements or vertical profiles alone. This approach enables insights into turbulent variability in complex terrain. Some remarks have to be made on the turbulence retrievals with Doppler lidars:

– The uncertainty of the measurement of LOS variance depends on the signal strength of the atmospheric backscatter. While the atmospheric conditions were sufficient for the data presented here, the availability of lidar measurements is limited in conditions of very low aerosol in the ABL, rain, or fog.

– An averaging of measured LOS velocity along the lidar beam is inherent to the Doppler lidar technology. The size of this averaging volume defines the limits of detectable eddies in the LOS velocities. For the systems used in this study, the width of this averaging window is on the order of a few tens of meters. For integral length scales $L_v$ smaller than this averaging window, an accurate measurement of LOS wind speed variance $\hat{\sigma}_v$ is not possible because the inertial subrange cannot be resolved. This limitation affects both the vertical stare measurements and the RHI measurements, as they both depend on LOS velocity variances.

– The analytic uncertainty estimation and the comparison to a sonic anemometer both show that the dissipation rates below $10^{-4}\ \mathrm{m^2 s^{-3}}$ cannot be resolved appropriately with lidar systems with the given physical resolution. Measurements below $10^{-3}\ \mathrm{m^2 s^{-3}}$ are already subject to high uncertainties.

Given these limitations, we still see that lidar measurements reliably detect time periods and spatial regions of increased turbulence.

Regarding the method of turbulence retrieval from RHI measurements, we find that:

– The size of sub-areas in the RHI plane as well as the averaging period should be chosen carefully depending on the scanning parameters (i.e. angular resolution, range gate distance, scan duration). In our case, a 20x20 $\mathrm{m^2}$ area and 30 minute averaging period was sufficient to capture the large turbulent scales for the calculation of the LOS variance. Large errors can be introduced if the measured LOS variance $\hat{\sigma}_v^2$ does not include the whole inertial subrange at large values of $L_v$.

– A careful calibration of $\sigma_0^2$ and $\Delta z$ with respect to reference instruments is necessary in order to obtain reliable results for TKE dissipation rate.

– While storing raw Doppler spectrum data of the wind lidars may seem expensive in a field campaign, these data enable averaging multiple measurements in the spectral domain and thus increasing the signal-to-noise ratio. Whenever possible, these raw spectra should be saved from field campaigns with interest in turbulence variability.

Given numerous logistical constraints, the lidars, sonics and TLS in this experiment could not collect measurements at the same location. Given the complex terrain, a difference in measurement location of only a few hundred meters can cause significant differences in the observations of the flow field and its turbulence. This spatial heterogeneity should be considered in the evaluation of the magnitude of correlation between the instruments that was found to be on the order of $R = 0.6$ in this



study. Previous studies evaluating profiling lidars with the same retrieval methods as presented in this study in flat terrain yield correlation coefficients larger than 0.7 in stable atmospheric conditions (Bodini et al., 2018). For future campaigns in complex terrain, co-located in-situ instruments in the measurement volume of the lidar are recommended to decrease the influence of spatial variability and improve the possibilities to validate lidar turbulence retrievals.

## 6.2   Wind turbine wakes in complex terrain

Menke et al. (2018) and Wildmann et al. (2018a) showed that wind turbine wakes in stable stratification at the Perdigão campaign can be observed far downstream, following the terrain into the valley of Vale do Cobrão. By means of wake tracking
algorithms based on the wind speed deficit, the wake could be detected up to ten rotor diameters downstream in very stable atmospheric conditions. From the RHI measurements shown in Figs. 12 and 16 increased turbulence is observed in the near-field of the WT and at least three rotor diameters downstream. To be able to distinguish wake induced turbulence from the background turbulence further downstream in the valley, it was necessary to include other observations of turbulence as well as information about wind speed, wind direction and potential temperature. Only the aggregate of all observations provides the
strong evidence that the wake of the wind turbine in the night of 14 June is trapped under the nighttime inversion and advected and stretched with the mean wind into the valley. Small wind direction changes cause large changes in observed turbulence at the specific instrument locations, which suggests that even at $11\,D$ downstream, the wind turbine wake is a local feature which is not completely eroded in the background turbulence. These data cannot quantify how much the background turbulence is affected by the wake or how turbulent mixing in the valley is enhanced by the presence of the wake. Extending the dataset to
more cases during the Perdigão campaign and a comparison to measurements at other locations and campaigns is necessary for conclusive analyses towards these goals.

## 7   Conclusions and outlook

We employ several instruments and analysis methods to provide a comprehensive view of turbulence structures and variability at the Perdigão 2017 field campaign. We quantify turbulence dissipation rate using vertically profiling lidars and a new anal-
ysis method using RHI lidar scans. These remote sensing methods compare well to traditional in situ methods, using sonic anemometers on meteorological towers or hot wire anemometers mounted on a tethered lifting system. We also offer means to quantify the uncertainty in dissipation rate estimates. For one case study, we find brief periods of disagreement between the methods, but we can attribute that disagreement to the propagation and meandering of a wind turbine wake which does not affect all measurements simultaneously.
This study gives a good example of the multitude and variety of methods and instruments that are available and beneficial to sample the complex flow in mountainous terrain. Within its limitations, lidar remote sensing is a powerful tool to sample wind and turbulence and provide spatio-temporal data which can be directly compared to numerical models. Utilizing the methods introduced in this study, more measurements by at least eight other lidar instruments performing RHI scans at the Perdigão 2017 campaign could be analyzed in future to expand the analysis of spatial distribution of turbulence and thus providing a





unique dataset for validation of numerical models in complex terrain.

A remaining challenge is the adequate sampling of very low turbulence in the stable ABL which cannot easily be improved with the current state of the art of lidar technology. A different kind of lidar systems or other measurement technology is necessary in these cases. Remotely piloted aircraft (RPA) are increasingly used in stable ABL research (Kral et al., 2018) as well as for investigations in complex terrain (Wildmann et al., 2017). As such, they are a promising tool to validate and complement remote-sensing data in similar ways as shown in this study.

The physics of WT wakes remains an important field of research for wind farm design and control. Providing spatial information of wind and turbulence with lidar is already and will still be of great importance for future research in the field. The
5    methods presented in this study can therefore not only provide valuable information about turbulence in complex terrain but also about turbulence in the wake of wind farms including offshore sites where wake effects can have a large impact on the mixing of the ABL in specific atmosperic conditions as observed in measurements (Platis et al., 2017) and meso-scale simulations (Siedersleben et al., 2018).

*Data availability.*  High-rate data from sonic anemometers on the meteorological masts (UCAR/NCAR, 2019) and quality-controlled ra-
10    diosonde data (UCAR/NCAR, 2018) as well as OU CLAMPS lidar data (Klein and Bell, 2017) is available through the EOL project website at www.eol.ucar.edu/field_projects/perdigao. DLR lidar data is available through https://perdigao.fe.up.pt/.



## Appendix A: Nomenclature

| | |
|---|---|
| $\varepsilon$ | TKE dissipation rate |
| $\varepsilon_r$ | dissipation rate estimations from lidar RHI |
| $\varepsilon_s$ | dissipation rate derived from sonic anemometer measurements |
| $\varepsilon_t$ | dissipation rate derived from TLS measurements |
| $\varepsilon_v$ | dissipation rate estimations from lidar vertical stare |
| $\sigma_{\varepsilon,s}$ | uncertainty of dissipation rate estimations by sonic anemometers |
| $\sigma_{\varepsilon,t}$ | uncertainty of dissipation rate estimations by TLS |
| $\sigma_{\varepsilon,v}$ | uncertainty of dissipation rate estimations by vertical stare lidar |
| $\sigma_{\varepsilon,r}$ | uncertainty of dissipation rate estimations by RHI lidar |
| $\sigma_{\sigma,v}$ | uncertainty of LOS velocity variance |
| $\sigma_{\sigma,t}$ | uncertainty of turbulent broadening |
| $\sigma_{\hat{\sigma},t}$ | uncertainty of lidar LOS velocity variance measurement |
| $\sigma_0^2$ | Doppler spectral width at zero wind speed |
| $\sigma_e^2$ | lidar instrumental noise |
| $\sigma_p$ | lidar pulse width |
| $\hat{\sigma}_s^2$ | lidar measured shear contribution to variance |
| $\hat{\sigma}_{\mathrm{sw}}^2$ | lidar measured Doppler spectral width |
| $\sigma_t^2$ | turbulent broadening of the Doppler spectrum |
| $\sigma_v^2$ | velocity variance |
| $\hat{\sigma}_v^2$ | lidar measured LOS velocity variance |
| $\sigma_{L,v}$ | uncertainty of integral length scale estimation |
| $\Delta R$ | distance between neighbouring range gate centers |
| $\Delta z$ | length of the lidar sensing volume |
| $\hat{v}_r$ | lidar measured LOS velocity |
| $\overline{v}_r$ | average lidar measured LOS velocity |
| $B_v$ | correlation function of flow velocity |
| $C_k$ | Kolmogorov constant |
| $D_v$ | structure function of velocity |
| $E$ | random error of spectral width measurement |
| $E_S$ | mean error between measured spectrum and model |
| $H_p$ | low-pass filter function for lidar measurement |
| $L_v$ | integral length scale |
| $Q_s$ | lidar sensing volume window function |
| $S_v$ | energy spectrum of flow velocity |
| $S_{\hat{v}}$ | lidar measured spectral energy |
| $T_w$ | lidar time window |





**Appendix B: Atmospheric conditions**

15  To understand the flow system with a LLJ from South-West which occurred in the night from 13 June to 14 June 2017, the long-term Weather Research and Forecasting model (WRF) simulation as described in Wagner et al. (2019a) is consulted. The meteorological situation was characterized by a synoptic low pressure system at 850 hPa, which was located over the Atlantic Ocean SW of the Iberian Peninsula. The combination of synoptic and thermally driven forcings and the interaction with the complex terrain around Perdigão resulted in a highly complex boundary-layer flow. Unlike in most nights during the Perdigão

5  2017 campaign a LLJ from southwest developed instead of the usual north-easterly LLJ (as it was also observed in the nights before and after this case study). Figure B1 shows the vertical profile of wind speed and wind direction up to a height of 6 km at the location of tower 20/trSE04. In the supplementary material, we provide maps of wind speed at 600 m above sea level, 850 hPa and 500 hPa, as well as a Hovmoeller plot from 11 June through 16 June 2019 to illustrate the synoptic situation during the night of the case study.

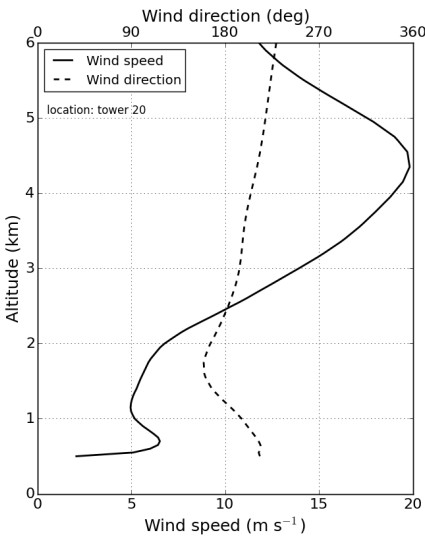

**Figure B1.** Wind speed and wind direction up to 6 km as an average from 0530-0600 UTC from WRF simulations.

10  **Appendix C: Spectral analysis of measured data**

For sonic anemometer, TLS and lidar vertical stare measurements, power spectra of measured flow velocity can be calculated and show how these instruments resolve turbulence at different scales. Only a careful choice of the scales that are used to derive $\varepsilon$, as it is done in this study, allows a valid comparison.





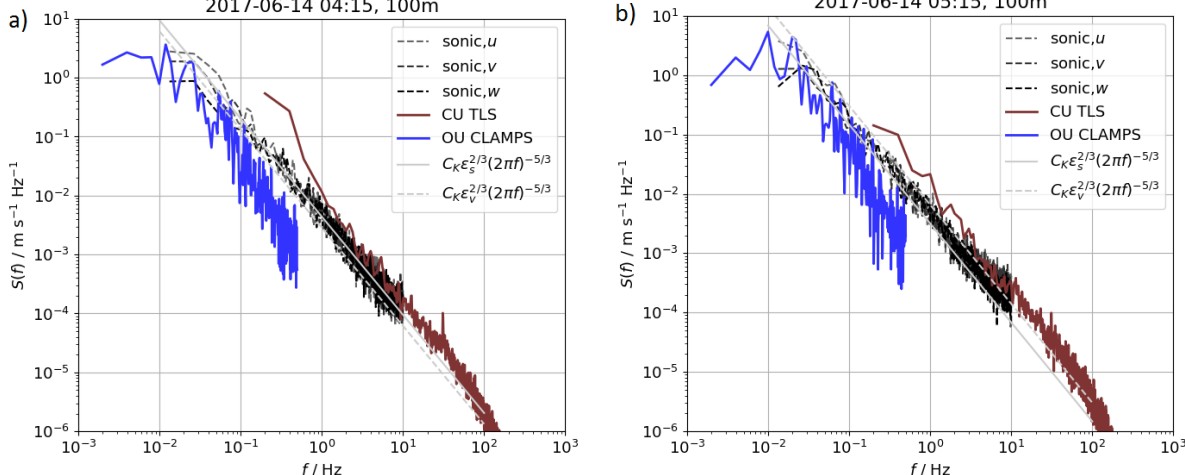

**Figure C1.** Variance spectrum of TLS (between 90 m and 110 m above the ground), vertical stare and sonic anemometer at 100 m above ground for the time period 0400-0430 UTC (a) and 0500-0530 UTC (b).

*Acknowledgements.* We want to thank José Palma, University of Porto and José Carlos Matos and the INEGI team for the local organization and tireless work in order to make this experiment a success. We acknowledge all the hard work of the DTU and NCAR staff to provide large

parts of the hardware and software infrastructure available at Perdigão. We also acknowledge Petra Klein and Tyler Bell for the measurements with the OU CLAMPS lidar and the provision of data.

We appreciate the hospitality and help we received from the municipality and residents of Alvaiade and Vila Velha de Rodão throughout the campaign.

We want to thank Anton Stephan and Thomas Gerz for internal review of the manuscript and their valuable comments.

This work was performed within projects LIPS and DFWind, both funded by the Federal Ministry of Economy and Energy on the basis of a resolution of the German Bundestag under the contract numbers 0325518 and 0325936A, respectively.

JKL and NB were supported by the US National Science Foundation CAREER Award 267 (AGS-1554055); JKL, NB, and LB were supported by the US National Science Foundation award AGS-1565498.





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
