# Peer review of "Estimation of turbulence dissipation rate from Doppler wind lidars and in-situ instrumentation in the Perdigão 2017 campaign"

_Atmospheric Measurement Techniques, 2019_

## Referee Comment (RC1)

**Review of "Estimation of turbulence parameters from scanning lidars and in-situ instrumenation in the Perdigão 2017 campaign" by Wildmann et al. 2019 (amt-2019-171)**

May 22, 2019

In this study, the authors propose a new approach to estimate TKE from RHI scans performed with Doppler lidars and compare the results to TKE retrieved from a vertical staring Doppler lidar and in situ measurements on towers and a tethered lifting system. The method is applied to data from the Perdigão field campaign which was conducted in Portugal in 2017. Finally, the authors present an example on how TKE from different instruments allows to investigate the wake of a wind turbine.

With the recent enhancements in remote sensing technology, undreamed-of possibilities to study flow structures and turbulence in the ABL appear. To my knowledge, this nice study presents a novel technique related to this topic and falls into the scope of AMT. The manuscript is overall well written and carefully prepared, relevant literature is cited and figures are of good quality. Nevertheless, I have several specific and two general comments - one is about the presentation of the technique to retrieve the dissipation rate from RHI scans and the other one is about the structure of the result sections. I believe that the manuscript is suitable for publication provided that the authors prepare a revised version of the manuscript considering my comments and suggestions.

**1 General comments**

1. Sect. 3.2.4: The method to derive dissipation rate from RHI is based on the idea of Smalikho et al. (2005). Currently, I find it hard to to distinguish between results of Smalikho et al. (2005) and modifications done by the authors. Where does the modification start? On p. 10, l. 17? I suggest making this much more clear in the text.

2. Sect. 4 and 5: The structure of these two sections is not straightforward and clear in my opinion. The authors start with some statistics of point-to-point comparison in Sect. 4.1,

but already introduce the case study which they will discuss later on (p. 16, l. 9ff). In Sect. 4.2 they first compare profiles from this specific case followed by some statistical comparison of profiles on p. 18, l. 9. In Sect. 5, they finally explain the wake during the case study in detail. I suggest re-arranging these two sections. For example, the authors could start with a section on the statistical point-to-point and profile comparisons for all available days (so basically the results of Fig. 6, 10 and 11. In a second section they could focus on the case study, first starting with the evolution of conditions throughout the night (Fig. 8) followed by the analysis of the wake. When re-arranging many of the figures could (should) be combined. For example, the profiles in Figs. 8 and 14, RHI scans of dissipation rates in Figs. 12 and 16 and coplanar wind in Figs. 7 and 13. This would also reduce the quite large number of figures. Also, the authors should consider to show corresponding times. For example, it would be interesting to see Fig. 13 at the same times as Fig. 12 and 16.

**2  Specific comments**

1. Title: As the authors focus on the dissipation rate, turbulence parameters could be replaced by dissipation rate.

2. Abstract: The abbreviations TKE, ABL, TLS, RHI, DSW, LOS and LLJ are not necessary in the abstract as they are not reused in the abstract.

3. Introduction: The structure of the introduction is not clear. In the current version, the authors start with turbulence over complex terrain, followed by model considerations and observations in general. While the proposed method can be applied to all kind of terrains, the data shown here are for complex terrain which imposes additional complexity. The authors could rearrange the introduction as follows: start with the challenges in models (wrong parameterization, impact on wind, importance for wind energy sector), continue with the methods to derive turbulence from observations and end with the particular issues related to complex terrain (which is presented in this study). This would then be more from general to specific. Also, the research objectives and motivation should be more clearly formulated and specified.

4. p. 2, l. 10: What does XPIA stand for?

5. p. 2, l. 15-16: This is especially true above the surface layer when few large convective cells can dominate the spectrum. A relevant study to cite in this context is Maurer et al. (2016).

6. p. 2, l. 20: RHI scans have been used as well to derive turbulence characteristics (Bonin et al., 2017).

7. p. 2, l. 26: Remove turbulence before TKE.

8. p. 2, l. 34: The content of Sect. 6 is not mentioned here.

9. p. 3, l. 6: It would be good to mention how far the ridges are apart.

10. p. 3, l. 23: The title reads scanning lidars. While this is true it is misleading, as the Halo Streamline is not used in scanning mode. I suggest rephrasing the title to Doppler lidars.

11. p. 3, l. 32: What is meant by flexible multi-Doppler lidar measurements?

12. Fig. 1: The rotation of the map is confusing in combination with the wind directions discussed in the result section. It would be good to add a small map to show where in Portugal the measurements were done. The contour labels are hard to read and the grey structures are not very clear to see. For #3, 4 RHI directions are given. Please indicate which ones are used in the analysis (Fig. 15). In general, the abbreviations used for the different sites are not consistent throughout the manuscript. For example, trSE_04 is named 20/trSE_04 and CU TLS is just labeled TLS in the text. The OU CLAMPS lidar is referred to as Halo Sreamline lidar in the caption of Table 2. This is unnecessarily confusing.

13. Fig. 2: I am confused by the position of lidar DLR #2. From Fig. 1 I have the impression that it is in the valley while in Fig. 2 it seems like it is on the slope. "lidar CLAMPS" should be labelled the same way as in Fig. 1.

14. Table 1: Please comment why the range date distances were chosen differently and give the physical resolution which results from the pulse length.

15. Table 2: What is the physical resolution of the Halo lidar?

16. p. 5, l. 3 and l. 10: Please comment a little more on the chosen specifications for the lidars. How continnous were the vertical stare measurements of the OU CLAMPS lidar? Where they only interrupted by the VADs every 15 min? You should also mention here that the Doppler spectra are stored during operation as they were needed for the analysis. How are the data filtered? Did the authors apply a SNR filter to detect erroneous radial velocity data before the LOS variance is calculated (Eq. 13)?

17. p. 7, l. 9: Please clarify what is meant by ensemble average in this context.

18. p. 7, l. 18: What does the integral length scale describe?

19. Sect 3.2: The order in this section (sonic, TLS, lidars) does not agree with the order in Sect. 2.2 (lidar, TLS, sonic).

20. p. 8, l. 7: Why not include the inertial dissipation technique in Sect 3.1?

21. p. 8, l. 15: No brackets around O'Connor et al. and Bodini et al.

22. p. 8, l. 17: How is the range of the inertial subrange determined.

23. p. 8, l. 23: Which height is used for U? One value for all heights or height dependent? It could be mentioned that the horizontal wind speed is not necessarily the propagation speed of

cells and the turbulence characteristics such as the integral length scale may be very sensitive to the choice of U. A recent study dealing with this topic is e.g. Adler et al. (2019).

24. p. 9, l. 1: "..measured lidar spectrum.."

25. p. 9, l. 17: How many different points are in one square sub-area?

26. p. 9, l. 20-21: Although 30 min are a common averaging intervals it might be too short and contributions by larger cells might be missed, in particular above the surface layer.

27. p. 11, l. 2: Eq. 13 and Eq. 20 both give expression for the measured velocity variance. Which one is used?

28. p. 11, l. 4: Please explain a little more what is shown in Fig. 3.

29. p. 11, l. 5: Please refer to Eq. 8 for the von Kármán model.

30. p. 12, l. 1: Under which assumptions is the equation for $H_p$ simplified?

31. p. 12, l. 5-9: Are these equations from Smalikho et al. (2005)? Why $\Delta_z$ and not $\Delta z$ in Eq. 29 and 30?

32. p. 12, l. 14: Why $\hat{L}_v$ and not $L_v$?

33. p. 12, l. 20: Why not $\sigma_v^2$? I thought the hat denotes measured variables?

34. Sect. 3.3.2 and Sect. 3.3.3: Are these methods new or have they been done before? Please cite appropriate literature or make it clear that this is novel.

35. p. 15, l. 12: ".. measurements are taken (Fig. 1)."

36. p. 15, l. 14: How is the interruption by VAD every 15 min affecting the retrieval of dissipation rate from vertial stare mode for 30 min intervals?

37. p. 16, l. 1-3: Here, the authors say that the scatter is related to the spatial separation. On p. 15, l. 12-13, they state that they expect a similar behavior. Please rephrase.

38. p. 16, l. 3: Where is shown that the uncertainty of the retrievals for vertical stare mode increases for weaker turbulence?

39. Fig. 6: How many data are used for the scatter plots? The squares and circles for DLR #2 and DLR #1 are hard to distinguish and the color range for low probability density is hard to see as well. Please change.

40. p. 16, l. 9 - p. 18, l. 7: This better fits to the case study section (see major comment 2).

41. Fig. 7: It would be more helpful for the analysis to show the same time periods as in Fig. 9.

42. p. 17, l. 5: Better say "..retrieved values of $\epsilon$..."

43. Fig. 8: It would be helpful to indicate the time period of the profiles and of the RHI scans shown in the other figures for the case study in the time series plot. Which bin is shown for the RHI scans and vertical stare? The ones closest to the tower measurement height? The plots acutally start at 21 UTC and not at 00 UTC as indicated in the caption.

44. p. 18, l. 3: "... at 0400-0430.."

45. p. 18, l. 6: I mainly see a better agreement above 400 m and not above the ridge.

46. p. 18, l. 9: In the caption of Fig. 10 it is stated that the period from 9-15 June is shown and in the text is says 14 June.

47. p. 19, l. 1: How are the values interpolated? Linearly?

48. Fig. 10: What are the number of values used? Figs should not be in the middle of the page surrounded by text, but rather at the top or bottom.

49. p. 19, l. 3: What is meant by these directions differ significantly? As the azimuth is the same, do the authors refer to the elevation angle? Why do the authors say that the dissipation rate should not depend on the direction? If the elevation angles of the lidars are different, they average over different volumes even when volume center is identical. So, they can be (slightly) different.

50. Fig. 11: Why not show this comparison for all available days? This should then much better fit to the (proposed) statistic section.

51. Sect. 5.1: While in the previous section mainly time intervals (e.g. 0500-0530 UTC) were given, the authors now use the center (?) times of the intervals (e.g. 0515 UTC). Please homogenize.

52. p. 20, l. 18: The information that there were radiosoundings and information on the launch time, etc should be given much earlier in the manuscript.

53. Fig. 12: Why do the authors not include the dissipation rates from the towers in these figures? Indicate in the caption what the squares and circles are.

54. p.22, l. 1: Where is the Lower Orange Site? Indicate in Fig. 1 and use uniform namings.

55. p. 22, l. 3: What time was sunrise? Are there temperature profiles available throughout the night? Was the whole valley filled by an inversion or was there a capping inversion near ridge height?

56. p. 22, l. 6: ".. at 235° (Fig. 8)."

57. p. 22, l. 14: I do not know what the authors mean by ring of large $\sigma_t^2$. Please indicate in

Fig. 15.

58. p. 22, l. 19-20: I am not sure I can follow this: The inversion is about 100 m above ridge height (from launches in the valley center). The high dissipation rates are confined to a layer below the ridge height. So how can they be trapped under the inversion?

59. Fig. 15: This figure is difficult to understand. It is hard to imagine where the slices are placed. It would be good to at a sketch here or in Fig. 1 to show where the slices are. Maybe a 3D plot would help as well, where the shown slices are placed in the 3D orography. Maybe the authors could even combine several slices from DLR #3 or even with the slices from # 1 and #2 to give a better impression of the 3D conditions.

60. Sect. 6.1: As this section mainly deals with turbulence measurements by Doppler lidar this should be specified in the title.

61. p. 24, l. 7-12: Isn't the variability within the averaging volume partly considered in $\sigma_t$?

62. p. 24, l. 13-15: The large uncertainties for small dissipation rates are likely related to the accuracy of the Doppler lidars.

63. p. 24, l. 21: "... was sufficient to capture..". How was this tested?

64. p. 24, l. 24-25: This is certainly a limitation for the method. Very often there is no option to calibrate with in situ measurements. Can the authors comment on how useful the method is, when no calibration can be performed?

65. p. 25, l. 10: How far downstream in terms of m are three rotor diameters?

66. p. 25, l. 25: Strictly speaking, the authors analyzed only on period in detail and attributed the disagreement to the wake. However, there are several more periods, when the dissipation rates from the different instrument differ.

67. p. 25, l. 29: "Within its limitations (Sect. 6.1)..."

68. p. 27: The list in Appendix A is not complete. For example, $\sigma_I$, $\sigma_e$, $\hat{L}_v$ are missing. Please complete.

69. Fig. B1: Why did the authors use WRF simulation for the large scale conditions when a radiosounding was available? What does "location: tower 20 mean" in the plot?

**References**

Adler, B., Kiseleva, O., Kalthoff, N., and Wieser, A.: Comparison of Convective Boundary-layer Characteristics from Aircraft and Wind Lidar Observations, Journal of Atmospheric and Oceanic Technology, doi:10.1175/JTECH-D-18-0118.1, 2019.

Bonin, T. A., Choukulkar, A., Brewer, W. A., Sandberg, S. P., Weickmann, A. M., Pichugina, Y. L., Banta, R. M., Oncley, S. P., and Wolfe, D. E.: Evaluation of turbulence measurement techniques from a single Doppler lidar, 10, 3021–3039, 2017.

Maurer, V., Kalthoff, N., Wieser, A., Kohler, M., Mauder, M., and Gantner, L.: Observed spatiotemporal variability of boundary-layer turbulence over flat, heterogeneous terrain, Atmos. Chem. Phys., 16, 1377–1400, 2016.

Smalikho, I., Köpp, F., and Rahm, S.: Measurement of atmospheric turbulence by 2-$\mu$ m Doppler lidar, J. Atmos. Oceanic Technol., 22, 1733–1747, 2005.

---

## Referee Comment (RC2) · Anonymous Referee #1 · 12 Jul 2019

The paper describes a new methodology for retrieving dissipation rate estimates from scanning Doppler Lidar RHI scans. The results are compared against a sonic anemometer, another vertical staring Lidar and TLS measurements. Overall the paper is well written and provide results that are consistent with theory, but there are some aspects which needs more clarity and further analysis. The reviewer has given some major and minor comments for the authors to consider in revising their article.

Major Comments: 1. Equation 19 should include the effects of Lidar Instrumental noise in this analysis. This has shown to significantly corrupt the Lidar data in many instances (Frehlich et al., 2006, Newsom et al., 2017). Please take a look at Lenschow

et al., 2000 and address that issue in your estimates from Lidar data. This could explain a lot of the variability the authors are seeing in low dissipation rate estimates. 2. Equation 19 should also include the covariance between the measured variance and turbulent broadening of the spectra. They are related and it needs to be accounted for in the equations. Please take that into account in your analysis. 3. Also, show the length scale estimates from the RHI Lidar retrievals compared to Sonic measurements. In low dissipation rate conditions, the uncertainty of this type of retrieval is high and this has a lot to do with proper length scale estimation. 4. Since Turbulence is more a statistical quantity, instantaneous snapshots of turbulence are not extremely helpful in decoding the trends within the atmosphere. So please show the below two plots in your analysis (See Shupe et al., 2012) a. Spectra of the Lidar and sonic measurements needs to show -5/3 and that you are able to resolve the inertial subrange with your measurements. b. Please show distributions of percent error between Lidar and tower measurements, since turbulence is a statistical quantity. Its important to understand how well the Lidar is doing for all conditions. 5. What time period of data was used for the Sonic calibration? 6. Please also state the expected performance of this algorithm in orthogonal wind directions. Looks like those were the cases, the measurements diverged significantly? 7. Figure 15 needs some imagination to confer with authors view, as the result is mostly noisy. I would recommend removing that figure and probably show a vertical profile of wind direction within the valley from one of the remote sensors? 8. Figures 12 & 16, although show the dissipation rate within the wake and the trapping of the turbulence within the valley as authors suggest but is extremely choppy. Maybe the height of the measurements can be limited to 200 m AGL for some clarity? 9. Since the authors had done Wake tracking in an earlier paper, can they provide a plot showing the decay of wake induced dissipation rate downwind from the turbine from these results? That would really add value to the paper. 10. Page 24: It is important to note, that the general remarks about turbulence retrievals with Doppler Lidars, especially second point about resolving length scales smaller than the rangegate is incorrect. Please see Frehlich et al., 2006, where Length scales smaller than

the range-gate size can be estimated using the azimuth structure function method. 11. There is a risk of the paper being too long, so I would recommend the authors to use the supplement section wisely to transfer some information into that section for brevity of the paper. Since most of the math is very similar to Smalikho et al, 2005, it would be recommended to have most of the equations relating to that in the supplemental section.

Minor Comments: 1. Distance between Lidar data & Sonic measurements? I think within 20 m, but maybe mention it in the article. 2. Page 12 Line 16: Remove the double dots after "Lv = 3" and mention "Lv = 3 to 1000 m". 3. Page 18 Line 3: Remove the double dots after "At 0400" 4. Figure 11 can be moved to the supplemental section. The variance is too high, and probably the look directions are different which is causing the large spread in estimates. 5. Looks like there was a similar dissipation analysis comparison done in the recent WFIP2 study, Wilzack et al., 2019 and this should be mentioned in the article as both talk about complex terrain and Lidar comparison in the introduction. 6. The analysis in Shupe et al., 2012 is very similar, albeit for Cloud Radars, the authors are recommended to take a look at that article for some interesting details.

Further references for authors to consider adding and review: • Frehlich, R., Meillier, Y., Jensen, M. L., Balsley, B., & Sharman, R. (2006). Measurements of boundary layer profiles in an urban environment. Journal of applied meteorology and climatology, 45(6), 821-837. • Shupe, M. D., Brooks, I. M., and Canut, G.: Evaluation of turbulent dissipation rate retrievals from Doppler Cloud Radar, Atmos. Meas. Tech., 5, 1375-1385, https://doi.org/10.5194/amt-5-1375-2012, 2012. • Newsom R.K., W.A. Brewer, J.M. Wilczak, D. Wolfe, S.P. Oncley, and J.K. Lundquist. 2017. "Validating Precision Estimates in Horizontal Wind Measurements from a Doppler Lidar." Atmospheric Measurement Techniques 10, no. 3:1229-1240. PNNL-SA-121097. doi:10.5194/amt-10-1229-2017 • Wilczak, J. M., Stoelinga, M., Berg, L. K., Sharp, J., Draxl, C., McCaffrey, K., ... & Muradyan, P. (2019). The Second Wind Forecast Improvement

Project (WFIP2): Observational Field Campaign. Bulletin of the American Meteorological Society, (2019).

---

## Author Comment (AC2) · 9 Aug 2019

**Estimation of turbulence parameters from Doppler wind lidars and in-situ instrumentation in the Perdigão 2017 campaign**

Norman Wildmann[1], Nicola Bodini[2], Julie K. Lundquist[2,3], Ludovic Bariteau[4], and Johannes Wagner[1]

[1]Deutsches Zentrum für Luft- und Raumfahrt e.V., Institut für Physik der Atmosphäre, Oberpfaffenhofen, Germany
[2]Department of Atmospheric and Oceanic Sciences, University of Colorado Boulder, Boulder, Colorado, USA
[3]National Renewable Energy Laboratory, Golden, Colorado, USA
[4]Cooperative Institute for Research in the Environmental Sciences, University of Colorado Boulder, Boulder, Colorado, USA

**Correspondence:** Norman Wildmann (norman.wildmann@dlr.de)

**1 Author response**

We want to thank the two anonymous reviewers for their valuable feedback and valid points of criticism to our manuscript.

**1.1 RC2, General Comments**

1. *Equation 19 should include the effects of Lidar Instrumental noise in this analysis. This has shown to significantly corrupt the Lidar data in many instances (Frehlich et al., 2006, Newsom et al., 2017). Please take a look at Lenschow et al., 2000 and address that issue in your estimates from Lidar data. This could explain a lot of the variability the authors are seeing in low dissipation rate estimates.*

   We agree that noise is critical, especially for the low turbulence regimes. For the RHI retrieval, two different kinds of lidar noise can be considered in fact. First, lidar instrumental noise can be included as $E$ for the spectral width determination in Eq. 14. However, following the example by Smalikho et al. (2005) and introducing a noise threshold $n_{\mathrm{th}}$ to the processing of the Doppler spectra, we assumed this contribution to be negligible. Next, following the example of the vertical stare retrieval, we did include in the revised manuscript another term in Eq. 19 to consider the noise contribution to the LOS variance, following the method of Pearson et al. (2009). From the results, which are presented in the revised manuscript, we cannot see a significant change in the large variability at low dissipation rates estimates. This is probably because other sources of uncertainty, connected to the atmospheric variability and the limits of the assumptions made with regards to turbulence theory, especially in complex terrain, outweigh the instrumental noise contribution. In addition, since the estimation of the noise contribution to the variance itself is not straightforward and relies on assumptions as well (see Lenschow et al. (2000)), we think it is important to provide an uncertainty estimation as shown in Sect. 3. For the revised manuscript we have significantly sharpened the filtering for the RHI retrieval, excluding all data with uncertainties larger than the actual value of $\varepsilon$. In the future, refinements of the method with improved error models might be possible, but we do not think that the Perdigão dataset is an appropriate one for this kind of analysis due to the large uncertainties connected to the complexity of the atmospheric flow.

2. *Equation 19 should also include the covariance between the measured variance and turbulent broadening of the spectra. They are related and it needs to be accounted for in the equations. Please take that into account in your analysis.*

   Eq. 19 follows directly from Eqs. 20 and 21. The theory is consistent and not missing a covariance term in that context.

3. *Also, show the length scale estimates from the RHI Lidar retrievals compared to Sonic measurements. In low dissipation rate conditions, the uncertainty of this type of retrieval is high and this has a lot to do with proper length scale estimation.*

   Yes, this is definitely true and we agree that it makes sense to show integral length scale estimates, since the dissipation rates are derived from them. We have calculated the integral time scale from the sonics using the variance of horizontal velocity and dissipation rate estimates according to Eq. 33 of the manuscript in order to be consistent with the theory for the lidar retrievals:

$$L_v = \frac{1.972}{C_k^{3/2}} \frac{\sigma_v^3}{\varepsilon} \quad . \tag{1}$$

   Figure 1 compares the results of $L_v$ and $\varepsilon$ for the whole period from 9 to 15 June. As the reviewer suggests, length scale estimation is naturally prone to larger uncertainties, which manifest in large variations at short timescales even in the sonic anemometer estimates. Integral length scale retrievals for corresponding dissipation rate estimates smaller than $10^{-4}\,\mathrm{m^2\,s^{-3}}$ have been removed because of the large errors introduced through Eq. 1 at low values of $\varepsilon$ and are excluded from the lidar retrieval already through the applied filters.

   In general, the diurnal cycle of turbulence and the corresponding length scales are well-captured by both the lidar and the sonic anemometer; however, some large variations emerge. We will include this plot in the revised manuscript and discuss it accordingly.

4. *Since Turbulence is more a statistical quantity, instantaneous snapshots of turbulence are not extremely helpful in decoding the trends within the atmosphere. So please show the below two plots in your analysis (See Shupe et al., 2012) a. Spectra of the Lidar and sonic measurements needs to show -5/3 and that you are able to resolve the inertial subrange with your measurements. b. Please show distributions of percent error between Lidar and tower measurements, since turbulence is a statistical quantity. Its important to understand how well the Lidar is doing for all conditions.*

   We agree that turbulence is a statistical quantity, however we are not presenting snapshots, but the results from half-hour statistics, as common for turbulence measurements and representation in the ABL. Spectra of lidar and sonic measurements are shown in Appendix C, Fig. C1. We make a very detailed analysis of the uncertainties of the lidar retrieval and give scatterplots of the comparison of all systems to show how well the retrievals work in different conditions.

5. *What time period of data was used for the Sonic calibration?*

   All available data from 9 June through 15 June was used for the calibration in order to get the best possible statistics over all occurring situations.

[Figure]

**Figure 1.** Comparison of time series of $L_v$ (a) and $\varepsilon$ (b) for the period form 9 to 15 June 2017 for the two RHI lidars DLR#1 and DLR#2 and the sonic anemometer at 100 m on tower 25/trSE_09 in the valley.

6. *Please also state the expected performance of this algorithm in orthogonal wind directions. Looks like those were the cases, the measurements diverged significantly?*

   Isotropy of turbulence is part of the assumptions that are made for the lidar retrievals, as well as for the in-situ retrievals. Proving or quantifying its validity is a challenging problem which we can not fully address with the available dataset. We can however see from the comparison between the RHI retrievals of the two lidars that are pointing at the same point with different beam angles that there is no systematic difference between the two systems.

7. *Figure 15 needs some imagination to confer with authors view, as the result is mostly noisy. I would recommend removing that figure and probably show a vertical profile of wind direction within the valley from one of the remote sensors?*

   We agree that the figure is disputable in its present form. This is mainly due to the weak performance of lidar DLR#3

cause by some technical issues. For this reason we will remove the figure from the manuscript. Wind direction profiles in the valley are given in Fig. 14 of the manuscript for all available instruments.

8. *Figures 12 & 16, although show the dissipation rate within the wake and the trapping of the turbulence within the valley as authors suggest but is extremely choppy. Maybe the height of the measurements can be limited to 200 m AGL for some clarity?*

In Fig. 12 we believe that it is important to show the heights up to 800 m to be able to see the upper part of the LLJ and the increased turbulence there. For Fig. 16 and the analysis of the wind turbine wake we agree that a limit to 200 m makes a lot of sense and will change the figures in the revised manuscript accordingly.

9. *Since the authors had done Wake tracking in an earlier paper, can they provide a plot showing the decay of wake induced dissipation rate downwind from the turbine from these results? That would really add value to the paper.*

For the case that we analyze here, plotting the decay of dissipation rate downwind of the wind turbine in the lidar RHI plane would be misleading. Our analysis shows that wind direction at the turbine location is not aligned with the RHI plane, as only in the valley winds veer towards the RHI plane. Given this complex wind field and the complex shape of the wake (as indicated in Fig. 15 of the manuscript), showing dissipation rate from the RHI plane as a function of distance to the wind turbine would not show the actual decay of turbulence in the wake. Given the methodology that we develop and validate in this study our plan is to look more closely at many occurrences of wind turbine wakes that are more aligned with the RHI plane throughout the campaign in the future. This is however out of the scope of this manuscript.

10. *Page 24: It is important to note, that the general remarks about turbulence retrievals with Doppler Lidars, especially second point about resolving length scales smaller than the rangegate is incorrect. Please see Frehlich et al., 2006, where Length scales smaller than the range-gate size can be estimated using the azimuth structure function method.*

We explicitly say that the restriction is there for vertical stare and RHI measurements as they are processed in this manuscript. The lead author has also worked with VAD methods using the azimuth structure function as described in Stephan et al. (2018b) and referencing Smalikho and Banakh (2017). Despite the theoretical possibility to retrieve turbulence parameters for $L_v < \Delta z$, these studies show that the sensing volume does influence the azimuthal structure function and it should be considered in the used turbulence model to reduce the systematic error. If these corrections are applied, $L_v$ needs to be larger than $\Delta z$ for theory to hold. We believe that it is important to emphasize the limitations of lidar measurements with the specific methods that are applied for turbulence retrievals. In our study, these limitations yield high uncertainties in low turbulence conditions that are inherent to the limitations of the resolution of the lidars. We will rephrase the statement on page 24 and also mention Frehlich et al. (2006) accordingly.

11. *There is a risk of the paper being too long, so I would recommend the authors to use the supplement section wisely to transfer some information into that section for brevity of the paper. Since most of the math is very similar to Smalikho et al, 2005, it would be recommended to have most of the equations relating to that in the supplemental section.*

In many papers about lidar retrievals, equations are spread over many references and are comparably hard to follow. In this manuscript we wanted to give a complete and still concise description of the full methodology for the interested reader. We are afraid that pushing many of the equations to the supplement would contradict this goal. We also think that in the final, two-column layout of the manuscript, the equations will not take as much space.

**1.2 RC2, Specific Comments**

1. *Distance between Lidar data & Sonic measurements? I think within 20 m, but maybe mention it in the article.*
   We mention the distance of the sonic anemometers from the RHI plane and CLAMPS in Sect. 4.1 and it can also be seen from Fig. 1, but we will additionally mention it in Sect. 2.2.3 in the revised manuscript. The distance is 150 m. We are aware that this is a comparatively large distance, but assess and discuss it throughout the manuscript.

2. *Page 12 Line 16: Remove the double dots after "Lv = 3" and mention "Lv = 3 to 1000 m".*
   Although the style guide of Copernicus journals explicitly allows the notation with dots (three dots though...) we will change it in the revised manuscript.

3. *Page 18 Line 3: Remove the double dots after "At 0400"*
   As previous.

4. *Figure 11 can be moved to the supplemental section. The variance is too high, and probably the look directions are different which is causing the large spread in estimates.*
   We agree that a lot of scatter is shown in the plot. A main cause for this are bad estimates of LOS velocity and turbulent broadening that propagate into the turbulence retrievals. In many cases these are due to hard target reflections, for example by clouds in far distance that will corrupt the Doppler spectra. However, we think the plot is important to show that there are no systematic differences between the two lidars. In order to make the plot better we sharpened the filtering of data significantly for the revised manuscript. Estimates with uncertainties larger than the actual value of $\varepsilon$ are removed and CNR-filters of $-25\,dB$ at the low end and $5\,dB$ at the high end are applied. For Fig. 11, we removed all occurrences with a probability density below 0.02 to remove remaining outliers and make the plot easier to read.

5. *Looks like there was a similar dissipation analysis comparison done in the recent WFIP2 study, Wilzack et al., 2019 and this should be mentioned in the article as both talk about complex terrain and Lidar comparison in the introduction.*
   The Wilczak et al. 2019 paper was published online just about at the same time that we submitted our paper. We will add it in the revised manuscript.

6. *The analysis in Shupe et al., 2012 is very similar, albeit for Cloud Radars, the authors are recommended to take a look at that article for some interesting details.*
   Thanks for the information. The method that is used in this article is very close to the method we use for the retrievals from vertical stare lidar measurements. We will reference the paper in the revised manuscript.

[Figure]

**Figure 2.** Comparison of all estimates of $\varepsilon$ in the RHI plane from DLR#1 and DLR#2 from 9 June through 15 June. The color scale represents the probability of occurrence of a measurement point. The black line is the line of identity.

**References**

Lenschow, D. H., Wulfmeyer, V., and Senff, C.: Measuring Second- through Fourth-Order Moments in Noisy Data, Journal of Atmospheric and Oceanic Technology, 17, 1330–1347, https://doi.org/10.1175/1520-0426(2000)017<1330:MSTFOM>2.0.CO;2, 2000.

Pearson, G., Davies, F., and Collier, C.: An Analysis of the Performance of the UFAM Pulsed Doppler Lidar for Observing the Boundary

5    Layer, Journal of Atmospheric and Oceanic Technology, 26, 240–250, https://doi.org/10.1175/2008JTECHA1128.1, 2009.

Smalikho, I., Köpp, F., and Rahm, S.: Measurement of Atmospheric Turbulence by 2-$\mu$m Doppler Lidar, Journal of Atmospheric and Oceanic Technology, 22, 1733–1747, https://doi.org/10.1175/JTECH1815.1, 2005.

Smalikho, I. N. and Banakh, V. A.: Measurements of wind turbulence parameters by a conically scanning coherent Doppler lidar in the atmospheric boundary layer, Atmospheric Measurement Techniques, 10, 4191–4208, https://doi.org/10.5194/amt-10-4191-2017, 2017.

10   Anton Stephan, Norman Wildmann, and Igor N. Smalikho: Spatiotemporal visualization of wind turbulence from measurements by a windcube 200s lidar in the atmospheric boundary layer. volume 10833, pages 10833 – 10833 – 10, https://doi.org/10.1117/12.2504468, 2018b.

---

## Author Comment (AC1)

**Estimation of turbulence parameters from Doppler wind lidars and in-situ instrumentation in the Perdigão 2017 campaign**

Norman Wildmann1, Nicola Bodini2, Julie K. Lundquist2,3, Ludovic Bariteau4, and Johannes Wagner1

1Deutsches Zentrum für Luft- und Raumfahrt e.V., Institut für Physik der Atmosphäre, Oberpfaffenhofen, Germany 2Department of Atmospheric and Oceanic Sciences, University of Colorado Boulder, Boulder, Colorado, USA 3National Renewable Energy Laboratory, Golden, Colorado, USA

4Cooperative Institute for Research in the Environmental Sciences, University of Colorado Boulder, Boulder, Colorado, USA

Correspondence: Norman Wildmann (norman.wildmann@dlr.de)

**1 Author response**

5

We want to thank the two anonymous reviewers for their valuable feedback and valid points of criticism to our manuscript.

**1.1 RC1, General Comments**

- 1. Sect. 3.2.4: The method to derive dissipation rate from RHI is based on the idea of Smalikho et al. (2005). Currently, I
- find it hard to to distinguish between results of Smalikho et al.(2005) and modifications done by the authors. Where does the modification start? On p. 10,1. 17? I suggest making this much more clear in the text.

The modification is explained on p.9, 11.7-18. We will to emphasize this more in the revised manuscript.

- 2. Sect. 4 and 5: The structure of these two sections is not straightforward and clear in my opinion. The authors start with some statistics of point-to-point comparison in Sect. 4.1, but already introduce the case study which they will discuss later
- on (p. 16, l. 9ff). In Sect. 4.2 they first compare profiles from this specific case followed by some statistical comparison of profiles on p. 18, l. 9. In Sect. 5, they finally explain the wake during the case study in detail. I suggest re-arranging these two sections. For example, the authors could start with a section on the statistical point-to-point and profile comparisons for all available days (so basically the results of Fig. 6, 10 and 11. In a second section they could focus on the case study, first starting with the evolution of conditions throughout the night (Fig. 8) followed by the analysis of the wake. When re-arranging many of the figures could (should) be combined. For example, the profiles in Figs. 8 and 14, RHI scans of
  - dissipation rates in Figs. 12 and 16 and coplanar wind in Figs. 7 and 13. This would also reduce the quite large number of figures. Also, the authors should consider to show corresponding times. For example, it would be interesting to see Fig. 13 at the same times as Fig. 12 and 16.

We

20

We have done major revisions to Sect. 4 and 5 according to the suggestions of the reviewer. Section 4 now contains only statistical analysis of all available data, whereas Sect. 5 describes the night of 13-14 June in detail, including the analysis

with regard to the wind turbine wake. Figures 7 and 13 as well as Fig. 12 and 16 have been combined. Figures 8 and 14 need to be kept separated in our opinion because Fig. 8 shows a time series while Fig. 14 shows vertical profiles.

**1.2 RC1, Specific Comments**

5

10

- 1. Title: As the authors focus on the dissipation rate, turbulence parameters could be replaced by dissipation rate.
- We agree that dissipation rate is the main focus of the study, so we will change the title accordingly.
  - 2. Abstract: The abbreviations TKE, ABL, TLS, RHI, DSW, LOS and LLJ are not necessary in the abstract as they are not reused in the abstract.

This is true, except for RHI, which is reused. We will eliminate the other abbreviations.

- 3. Introduction: The structure of the introduction is not clear. In the current version, the authors start with turbulence over complex terrain, followed by model considerations and observations in general. While the proposed method can be applied to all kind of terrains, the data shown here are for complex terrain which imposes additional complexity. The authors could rearrange the introduction as follows: start with the challenges in models (wrong parameterization, impact on wind, importance for wind energy sector), continue with the methods to derive turbulence from observations and end with the particular issues related to complex terrain (which is presented in this study). This would then be more
- 15 from general to specific. Also, the research objectives and motivation should be more clearly formulated and specified. We have revised and reorganized the introduction according to the suggestions by the reviewer. We now describe the challenges in models at the beginning of the introduction, but we think that we should still present the different mechanisms for turbulence generation before presenting the state-of-the-art of methods for turbulence retrievals from observations. Before giving the outline of the paper we have added a clear statement of the goals we pursue in this study.
- 4. p. 2, l. 15-16: This is especially true above the surface layer when few large convective cells can dominate the spectrum.
  A relevant study to cite in this context is Maurer et al. (2016).

We will add the mentioned study to the references.

- 5. *p.* 2, *l.* 20: *RHI scans have been used as well to derive turbulence characteristics (Bonin et al., 2017).* We will add the study and information about turbulence retrievals from RHI scans.
- 25 6. p. 2, l. 26: Remove turbulence before TKE.

Thanks, we have corrected this mistake in the revised manuscript.

7. p. 2, l. 34: The content of Sect. 6 is not mentioned here.

Section 6 deals with an assessment of data quality and results of the analysis. We will mention it in the revised manuscript.

- 8. p. 3, l. 6: It would be good to mention how far the ridges are apart.
- We will include an approximate distance between the ridges of 1400 m in the text in the revised manuscript.

30

9. p. 3, l. 23: The title reads scanning lidars. While this is true it is misleading, as the Halo Streamline is not used in scanning mode. I suggest rephrasing the title to Doppler lidars.

We agree that Doppler wind lidars is the more appropriate term in the title and will change it accordingly.

10. p. 3, l. 32: What is meant by flexible multi-Doppler lidar measurements?

- 5 We will eliminate the term 'flexible' because it does not contain any valuable information. It is mentioned in the text that the multi-Doppler measurements are made in a range of wind directions and more details are provided in the given reference.
  - 11. *Fig. 1: The rotation of the map is confusing in combination with the wind directions discussed in the result section.* We have changed the map to not be rotated in the revised manuscript.

**10 *It would be good to add a small map to show where in Portugal the measurements were done.**

We think that the references and the text give a good idea about the location of the experiment in Portugal and another map would overload this figure. It is also not really relevant for the study. For those readers interested in mesoscale and synoptic circulation, the supplementary material gives maps of the whole Iberian Peninsula, and indicate the experiment location in these.

**15 The contour labels are hard to read and the grey structures are not very clear to see.**

20

25

30

The contour label size will be increased in the revised manuscript. The grey structures shall just give an impression about the distribution of roughness elements.

For #3, 4 RHI directions are given. Please indicate which ones are used in the analysis (Fig. 15).

All of these directions are used for DLR#3. Each direction corresponds to a virtual tower in a different distance to the wind turbine as shown in Fig. 14.

In general, the abbreviations used for the different sites are not consistent throughout the manuscript. For example, trSE 04 is named 20/trSE 04 and CU TLS is just labeled TLS in the text. The OU CLAMPS lidar is referred to as Halo Sreamline lidar in the caption of Table 2. This is unnecessarily confusing.

- We will make sure the labels for instruments are consistent in the revised manuscript. Towers should be labelled according to the scheme 20/trSE\_04, the TLS will be labelled TLS consistently and the vertical staring lidar will be labelled CLAMPS.
  - 12. Fig. 2: I am confused by the position of lidar DLR#2. From Fig. 1 I have the impression that it is in the valley while in Fig. 2 it seems like it is on the slope. "lidar CLAMPS" should be labelled the same way as in Fig. 1.

The position of DLR#2 is consistent in all plots and in the photograph of Fig. 2. It is not at the lowest point in the valley in the given cross-section, as can also be seen in Fig.1 in the contour lines.

13. Table 1: Please comment why the range date distances were chosen differently and give the physical resolution which results from the pulse length.

Initially, scans with range gate distances of 10 m for all three DLR lidars were desired. However, in the course of the

3

campaign, it was desired to measure more of the inflow of the wind turbine, which can be captured with DLR#1. With a given limitation of the total number of range gates, the range gate distance was increased for this purpose in DLR#1. Since there was no reason to increase the range for DLR#2 and DLR#3, the range gate distance was kept as low as 10 m. The specified physical resolution by the manufacturer for the given configuration is 50 m and will be added to the table.

5

**14. Table 2: What is the physical resolution of the Halo lidar?**

The specified physical resolution of the CLAMPS lidar is 30 m. It will be added to Tab. 2 which will be merged into Tab. 1 in the revised manuscript.

15. p. 5, l. 3 and l. 10: Please comment a little more on the chosen specifications for the lidars. How continous were the

10 vertical stare measurements of the OU CLAMPS lidar? Where they only interrupted by the VADs every 15 min? You should also mention here that the Doppler spectra are stored during operation as they were needed for the analysis. How are the data filtered? Did the authors apply a SNR filter to detect erroneous radial velocity data before the LOS variance is calculated (Eq. 13)?

We will explain the measurement sequence of the CLAMPS lidar in more detail in the revised manuscript: nine minutes

15 of vertical stare measurements are followed by six minutes in which one along-valley RHI, one cross-valley RHI and one VAD scan are performed. For the CLAMPS lidar, only LOS data is used. Doppler-spectra are only used for the RHI retrieval. The CNR-threshold for CLAMPS was set to -23 dB. We will add this information in Sect. 3 where we think it is more suitable.

**16. p. 7, l. 9: Please clarify what is meant by ensemble average in this context.**

20

25

The ensemble average is the mean over n identical experiments. This is a theoretical, ideal case, which cannot be achieved in the atmosphere. Usually, spatial or temporal averages are used in experiments under the assumption of homogeneity (see Stull, 1988).

**17. p. 7, l. 18: What does the integral length scale describe?**

The integral length scale is precisely defined by Eq. 7. Integral length scales characterize the range of energy-containing eddy length scales.

- 18. Sect 3.2: The order in this section (sonic, TLS, lidars) does not agree with the order in Sect. 2.2 (lidar, TLS, sonic). We will change the order in Sect. 2.
- 19. p. 8, l. 7: Why not include the inertial dissipation technique in Sect 3.1?Because it is a technique that is used in this case for the TLS data and not a basic equation.
- 20. p. 8, l. 15: No brackets around O'Connor et al. and Bodini et al.
  Will be corrected in the revised manuscript.

**21. p. 8, l. 17: How is the range of the inertial subrange determined.**

A detailed description of the determination of the inertial subrange and N is given in Bodini et al. (2018). The reference will be added in the revised manuscript.

22. p. 8, l. 23: Which height is used for U? One value for all heights or height dependent? It could be mentioned that the

horizontal wind speed is not necessarily the propagation speed of cells and the turbulence characteristics such as the integral length scale may be very sensitive to the choice of U. A recent study dealing with this topic is e.g. Adler et al. (2019).

We use wind speed at the same height where dissipation is calculated. We are aware and it is mentioned in all references that the assumption of frozen turbulence is inherent to this method and a violation will lead to errors. This is however also the case for stationary in-situ measurements. Also, integral length scales are not determined from the vertical stares.

23. p. 9, l. 1: "..measured lidar spectrum.."

Will be changed in the revised manuscript.

**24. p. 9, l. 17: How many different points are in one square sub-area?**

This varies with distance from the lidar. Close to the lidar, there are as many as 15 points, while far from the lidar it can be reduced to 3-5 points. Most of the large scale variance is captured through the temporal average of 30 minutes and

5

10

15

20

be reduced to 3-5 points. Most of the large scale variance is captured through the temporal average of 30 minutes and tests showed that an increase or decrease of measurement points in the sub-areas does not change the results significantly.

25. p. 9, l. 20-21: Although 30 min are a common averaging intervals it might be too short and contributions by larger cells might be missed, in particular above the surface layer.

We agree that large cells and coherent structures are important contributions to turbulent mixing, but they are just as difficult to measure and more specific studies related to these phenomena will be necessary in the future to understand their impact on turbulence measurements with lidars.

26. p. 11, l. 2: Eq. 13 and Eq. 20 both give expression for the measured velocity variance. Which one is used?

Eq. 13 describes how velocity variance is calculated from measured data. Eq. 20 relates this to the theoretical model.

- 27. p. 11, l. 4: Please explain a little more what is shown in Fig. 3.
- Figure 3 is an illustration of the theoretical model for atmospheric turbulence and the theoretical lidar spectra that are measured with the lidar. We will give some more explanation in the revised manuscript.
  - 28. p. 11, l. 5: Please refer to Eq. 8 for the von K'arm'an model.

Will be added in the revised manuscript.

- 29. p. 12, l. 1: Under which assumptions is the equation for Hp simplified?
- 30 The simplified version assumes a Gaussian window shape. The information will be added to the manuscript.

**30.** p. 12, l. 5-9: Are these equations from Smalikho et al. (2005)? Why $\Delta_z$ and not $\Delta z$ in Eq. 29 and 30?**

Yes the equations are from Smalikho et al. 2005 as we say in the text. The  $\Delta_z$  is just a typo that will be corrected in the revised manuscript.

**31. *p.* 12, *l.* 14: Why $\hat{L}_v$ and not $L_v$ ?**

10

- 5 The hat is used to distinguish between the approximation and the analytic solution for  $L_v$ . Later in the text we mention that we will only use  $L_v$  for simplicity in the rest of the manuscript.
  - 32. p. 12, l. 20: Why not  $\sigma_v^2$ ? I thought the hat denotes measured variables?

We mean the total LOS velocity variance as desribed in Eq. 19 here, we will change the text accordingly.

33. Sect. 3.3.2 and Sect. 3.3.3: Are these methods new or have they been done before? Please cite appropriate literature or make it clear that this is novel.

These methods are basic error propagation applied to the equations in Sect. 3.2. We are not aware that this has been done exactly this way before and therefore we introduce it in Sect. 3.3.

- 34. p. 15, l. 12: ".. measurements are taken (Fig. 1)." Ok.
- 15 35. p. 15, l. 14: How is the interruption by VAD every 15 min affecting the retrieval of dissipation rate from vertial stare mode for 30 min intervals?

The algorithm is based on the small scales in the inertial subrange. If all assumptions hold and turbulence is stationary within the 30 minute interval, the retrieval should not be affected by the interruption, only the statistical error will increase.

20 36. p. 16, l. 1-3: Here, the authors say that the scatter is related to the spatial separation. On p. 15, l. 12-13, they state that they expect a similar behavior. Please rephrase.

Similar does not mean that exactly the same values are expected, so we do not see a contradiction in these statements. In any case, we will rephrase, stating that we expect similar diurnal development.

- 37. p. 16, l. 3: Where is shown that the uncertainty of the retrievals for vertical stare mode increases for weaker turbulence?
- It would be better expressed by saying that the relative uncertainty  $\frac{\sigma_{\varepsilon}}{\varepsilon}$  is higher for weaker turbulence. This follows from Eq. 37. In Fig. 1 we show the relative uncertainty in dependency of the LOS variance.

We think it is not necessary to show this plot in the manuscript, but will explain that the statement is based on Eq. 37.

- 38. Fig. 6: How many data are used for the scatter plots? The squares and circles for DLR#2 and DLR#1 are hard to distinguish and the color range for low probability density is hard to see as well. Please change.
- 30 As it is described in Sect. 4.1, RHI measurements with the described configuration are only available from 9-15 June. There are 149 valid half-hour periods for DLR#1 and 89 valid periods for DLR#2 after filtering has been applied. The

Figure 1. Relative uncertainty of dissipation rate retrieval from vertical stare measurements in dependency of LOS variance.

difference occurs due to the CNR of the lidar measurement at the location which is compared to the sonic anemometer. A lower CNR is more likely to be filtered. DLR#2 is situated rather close to the tower ( $\sim$ 120 m). With the far focus setting of the lidars ( $\sim$ 1000 m), the CNR at this point is significantly lower for this instrument compared to DLR#1 with a distance of  $\sim$ 500 m to the tower and thus, more data in low signal conditions are filtered.

The vertical stare lidar was operated in the same configuration from 6 May to 15 June and thus considerably more data is available with over 1400 valid half-hour periods.

We will modify the color map in the revised manuscript as in Fig. 2. It will be easier to distinguish individual points between DLR#1 and DLR#2 then, too. However, in the range of high point density it will still be difficult, but we do not consider this critical, because the results are very similar and would not justify a second plot. A comparison of the two lidars is given in Fig. 11.

39. p. 16, l. 9 - p. 18, l. 7: This better fits to the case study section (see major comment 2).

We did restructure the manuscript following the suggestions by the reviewer and the paragraph is moved to the case study section accordingly.

- 40. Fig. 7: It would be more helpful for the analysis to show the same time periods as in Fig. 9.
- 15

5

10

This will be changed in the revised manuscript. We will show the time period 0400-0430 UTC (see Fig. 3).

41. p. 17, l. 5: Better say "..retrieved values of  $\varepsilon$ ..."

It will be changed in the revised manuscript, although these statements are about sonic anemometer measurements and there is some discussion if the methods to obtain dissipation rate from sonic anemometers should be called a retrieval. Probably the best way is to call it an estimate.

Figure 2. Scatterplot of RHI dissipation retrievals compared to sonic anemometer estimates.

Figure 3. Coplanar wind field for the half-hour period 0400-0430 UTC.

- 42. Fig. 8: It would be helpful to indicate the time period of the profiles and of the RHI scans shown in the other figures for the case study in the time series plot. Which bin is shown for the RHI scans and vertical stare? The ones closest to the tower measurement height? The plots acutally start at 21 UTC and not at 00 UTC as indicated in the caption.
  - We are afraid that the plot that is already quite busy would be overloaded if indications of time periods were added to it. The time periods are given in all plots and should be easy to find in this plot as well. The caption has been corrected for the right time period in the revised manuscript.
- 5
- 43. p. 18, l. 3: "... at 0400-0430.."

Although the style guide of Copernicus journals explicitly allows the notation with dots (three dots though...) we will change it in the revised manuscript.

10 44. p. 18, l. 6: I mainly see a better agreement above 400 m and not above the ridge.

At 0400-0430 UTC at 50 m above the ridge, the difference between tower, CLAMPS and TLS is almost one order of

magnitude, whereas there is an almost perfect agreement between the systems from 0500-0530 UTC. The variability in TLS and DLR#1 and DLR#2 from 0400-0430 UTC is much higher between 0-200 m, compared to 0500-0530 UTC. For these reasons we think that our statement is justified.

- 45. p. 18, l. 9: In the caption of Fig. 10 it is stated that the period from 9-15 June is shown and in the text is says 14 June.
- "...for 14 June 2017" is a typo and will be corrected in the revised manuscript.
- 46. p. 19, l. 1: How are the values interpolated? Linearly?

5

10

15

20

Yes, we consider a linear interpolation in the rather high vertical resolution of the measurement points to be the most reasonable solution.

47. Fig. 10: What are the number of values used? Figs should not be in the middle of the page surrounded by text, but rather at the top or bottom.

In this scatter plot, the same time period as in Fig.6(b) is considered, but for height levels between -160 m and 800 m above ridge height. This yields more than 3500 points.

- 48. p. 19, l. 3: What is meant by these directions differ significantly? As the azimuth is the same, do the authors refer to the elevation angle? Why do the authors say that the dissipation rate should not depend on the direction? If the elevation
- angles of the lidars are different, they average over different volumes even when volume center is identical. So, they can be (slightly) different.

If the theory of local isotropy of turbulence at the small scales, which is the basis for the methods described in Sect. 3 holds, there should not be a systematic difference of dissipation rates retrieved with different elevation angles. Of course, small differences due to different sensing volumes are expected and manifest in the scatter of Fig. 11. We hope this explanation is clearer and will rephrases the revised manuscript accordingly.

49. Fig. 11: Why not show this comparison for all available days? This should then much better fit to the (proposed) statistic section.

We included all available days in the revised manuscript. The R-value increases that way, but the conclusions we can draw are not changing.

25 50. Sect. 5.1: While in the previous section mainly time intervals (e.g. 0500-0530 UTC) were given, the authors now use the center (?) times of the intervals (e.g. 0515 UTC). Please homogenize.

We will make sure to correct this in the revised manuscript.

- 51. p. 20, l. 18: The information that there were radiosoundings and information on the launch time, etc should be given much earlier in the manuscript.
- 30 We will add this information in Sect. 2.2.
  - 52. Fig. 12: Why do the authors not include the dissipation rates from the towers in these figures? Indicate in the caption what the squares and circles are.

In Fig. 12 we want to show the vertical structure and spatial distribution of dissipation rate. The tower measurements would overload this figure and be very hard to read. In the revised manuscript we zoom into the plot for the wake analysis and in these plots, tower measurements will be included.

53. p.22, l. 1: Where is the Lower Orange Site? Indicate in Fig. 1 and use uniform naming

- 5 The Lower Orange Site is introduced in Sect. 2.2.1 as the site where the CLAMPS lidar is placed. Since it is only mentioned at one other point in the manuscript, we will just refer to it as the CLAMPS site instead in the revised manuscript.
  - 54. p. 22, l. 3: What time was sunrise? Are there temperature profiles available throughout the night? Was the whole valley filled by an inversion or was there a capping inversion near ridge height?
- 10 Sunrise was at 0601 UTC. Unfortunately temperature is not sampled spatially as wind is in the Perdigão experiment. There are continuous vertical profiles of temperature by microwave radiometers, which are however known to not resolve the inversion layers very well. Numerical simulations could be used to study the thermal stratification in more detail, this is however beyond the scope of this manuscript and will have to be targeted in a future study.

55. p. 22, l. 6: ".. at 235° (Fig. 8)."

15

20

25

30

OK.

56. p. 22, l. 14: I do not know what the authors mean by ring of large  $\sigma_t^2$ . Please indicate in Fig. 15

In response to this comment and comments by reviewer #2 we will remove Fig. 15 from the manuscript. The data by lidar DLR#3 is poor due to some technical issues of the lidar instrument.

57. p. 22, l. 19-20: I am not sure I can follow this: The inversion is about 100 m above ridge height (from launches in the valley center). The high dissipation rates are confined to a layer below the ridge height. So how can they be trapped under the inversion?

We admit that this is a statement of an assumption we made but cannot proof adequately with the available dataset, especially because we do not have spatial data of temperature. We will rephrase this in the revised manuscript.

58. Fig. 15: This figure is difficult to understand. It is hard to imagine where the slices are placed. It would be good to at a sketch here or in Fig. 1 to show where the slices are. Maybe a 3D plot would help as well, where the shown slices are placed in the 3D orography. Maybe the authors could even combine several slices from DLR#3 or even with the slices from #1 and #2 to give a better impression of the 3D conditions.

The slices are indicated explicitly in Fig. 1, just like for the RHI plane of DLR#1 and DLR#2. We did combine the slices of all lidars to do a virtual tower retrieval, which is also shown in Fig. 14. Due to reasons mentioned above we have however decided to remove Fig. 15 from the manuscript.

59. Sect. 6.1: As this section mainly deals with turbulence measurements by Doppler lidar this should be specified in the title.

It is true that most of the specific points are for Doppler lidar measurements, however there are also general statements given that refer to all measurement instruments.

**60.** p. 24, l. 7-12: Isn't the variability within the averaging volume partly considered in $\sigma_t$ ?**

- Yes, the variability within the averaging volume is reflected in  $\sigma_t$ . For the method that is described in Sect. 3.2.4 to work and all assumptions to hold,  $\sigma_t$  needs to in fact only contain scales within the inertial subrange, which is not the case if  $L_v$  is smaller than the averaging window (see Fig. 3).
- 61. p. 24, l. 13-15: The large uncertainties for small dissipation rates are likely related to the accuracy of the Doppler lidars. As shown in the uncertainty propagation in Sect. 3.3, the uncertainty of dissipation rate is clearly depending on the uncertainty of the lidar variance measurements. However, the theory on which lidar retrievals are based has some limitations on integral length scales in dependency of lidar sensing volume as well which are described in this manuscript.

**62. p. 24. l. 21: "... was sufficient to capture..". How was this tested?**

We did calculate the results with smaller and larger areas and time periods. We agree that not having shown this in the analysis makes the statement hard to justify and we can also not guarantee that it is true for all situations and points in space. A time period of 30 minutes is however a common averaging period in ABL turbulence research. We will rephrase accordingly.

63. p. 24, l. 24-25: This is certainly a limitation for the method. Very often there is no option to calibrate with in situ measurements. Can the authors comment on how useful the method is, when no calibration can be performed?

We believe that we can show that for the case of the Perdigão campaign, the method is very helpful to show the spatial distribution of dissipation rates as it has never been shown before. An investment of one sonic anemometer for example with a ten meter mast is low compared to the lidar instrument and we would strongly recommend it. The method can also be applied without the calibration but some improved characterization of the lidar parameters themselves, for example in a laboratory, would be necessary. We believe there are a lot of prospects for enhancements and improvements to make the method more robust, even with a standalone instrument.

**64. p. 25, l. 10: How far downstream in terms of m are three rotor diameters?**

One rotor diameter is 82 m. This information will be added to the manuscript.

65. p. 25, l. 25: Strictly speaking, the authors analyzed only on period in detail and attributed the disagreement to the wake. However, there are several more periods, when the dissipation rates from the different instrument differ.

In fact, this period has the largest systematic disagreement between the instruments over a rather long period. It is introduced and presented as a case study of a very interesting phenomenon. Looking into other cases could be part of future work.

- 66. p. 25, l. 29: "Within its limitations (Sect. 6.1)..." Ok.
- 30

5

10

15

20

25

- 67. p. 27: The list in Appendix A is not complete. For example,  $\sigma_I$ ,  $\sigma_e$ ,  $\hat{L}_v$  are missing. Please complete.  $\sigma_e^2$  is given is given in the appendix. The other two variables will be added.
- 68. Fig. B1: Why did the authors use WRF simulation for the large scale conditions when a radiosounding was available? What does "location: tower 20 mean" in the plot?
- 5 Figure B1 goes in combination with the supplementary material which is provided for the interested reader to get more acquainted with the mesoscale situation and is less relevant for the study itself. Radiosoundings do not give information about the mesoscale and synoptic circulations as the WRF simulation does. We will add the radiosonde data to the plot to make the connection between simulation and measurements (Fig. 4).

---

## Referee Report (RR1)

**Second Review of "Estimation of turbulence parameters from scanning lidars and in-situ instrumenation in the Perdigão 2017 campaign" by Wildmann et al. 2019 (amt-2019-171)**

September 23, 2019

In this second version of the manuscript the authors included many of my comments from the first review and I think that in particular the rearrangement of the introduction and of the result sections 4 and 5 improved the clarity of the manuscript. I have few minor comments left. The page and line numbers refer to the manuscript version with the tracked changes in the authors' response.

1. p. 2, l. 15: What are turbulence models? It should rather be "turbulence parameterisation" or "numerical weather models".

2. p. 3, l. 16: What is a valley system? Do the authors mean a valley wind system?

3. p. 9, l. 3: The integral length scale describes the scale over which turbulence remains correlated (e.g. Kaimal and Finnigan, 1994). I suggest adding this verbal description.

4. p. 10, l. 27: In my first review, I asked for the number of point in the square sub-area (comment 24) to which the authors responded in their comments. However, I think that this information should be added to the manuscript as well, as it is helpful for the interested reader.

5. p. 10, l. 28: Like in the previous comment, the possible implication of the 30-min averaging intervals should be mentioned in the manuscript as well (comment 25 in the first review).

6. Fig. 6: As no data for CLAMPS are shown in (a), a legend should be plotted for each of the subplots only including the variables which are actually shown.

7. Fig. 7: The caption does not fit to (a) and (b). I believe (a) and (b) are switched, i.e. (a) is showing the results for the RHI and (b) the results for CLAMPS?

8. Figs. 7 and 8: In the captions it says "the color scale represent the density of probability of

a measurement point" (Fig. 7) and "the probability of occurrence of a measurement point" (Fig. 8). This should be uniform.

9. p. 21, l. 25 and Fig. 11: Maybe I am missing it, but I cannot find the information where the RHI profile of dissipation rate come from. Are they averages over same area across the valley or are they individual grid point values?

10. p. 25, l. 9: ".. and 0700 UTC (Fig. 12a).

11. p. 27, l. 1: "...wake induced turbulence being trapped under the inversion..." I don't see enough evidence for this in the data (comment 57 in the first review) and this should be rephrased.

**References**

Kaimal, J. C. and Finnigan, J. J.: Atmospheric boundary layer flows: their structure and measurement, Oxford university press, 1994.

---

## Author Response (AR2)

**Estimation of turbulence parameters from Doppler wind lidars and in-situ instrumentation in the Perdigão 2017 campaign**

Norman Wildmann[1], Nicola Bodini[2], Julie K. Lundquist[2,3], Ludovic Bariteau[4], and Johannes Wagner[1]

[1]Deutsches Zentrum für Luft- und Raumfahrt e.V., Institut für Physik der Atmosphäre, Oberpfaffenhofen, Germany
[2]Department of Atmospheric and Oceanic Sciences, University of Colorado Boulder, Boulder, Colorado, USA
[3]National Renewable Energy Laboratory, Golden, Colorado, USA
[4]Cooperative Institute for Research in the Environmental Sciences, University of Colorado Boulder, Boulder, Colorado, USA

**Correspondence:** Norman Wildmann (norman.wildmann@dlr.de)

**1 Review response**

We want to thank the reviewer for their valuable feedback to our manuscript.

**1.1 RC1, Comments**

1. *p. 2, l. 15: What are turbulence models? It should rather be "turbulence parameterisation" or "numerical weather models".*

   The parameterization of turbulence is often referred to as sub-grid scale models (see e.g. Deardorff, 1985). The text is changed to "sub-grid scale turbulence modeling".

2. *p. 3, l. 16: What is a valley system? Do the authors mean a valley wind system?*

   We changed the text to "valley flow".

3. *p. 9, l. 3: The integral length scale describes the scale over which turbulence remains correlated (e.g. Kaimal and Finnigan, 1994). I suggest adding this verbal description.*

   The sentence is added to the manuscript.

4. *p. 10, l. 27: In my first review, I asked for the number of point in the square sub-area (comment 24) to which the authors responded in their comments. However, I think that this information should be added to the manuscript as well, as it is helpful for the interested reader.*

   We added the information to the manuscript. The precise spatial distribution of measurement points can also be reproduced from the information given in Tab. 1.

5. *p. 10, l. 28: Like in the previous comment, the possible implication of the 30-min averaging intervals should be mentioned in the manuscript as well (comment 25 in the first review).*

   We incorporated our response into the manuscript.

6. *Fig. 6: As no data for CLAMPS are shown in (a), a legend should be plotted for each of the subplots only including the variables which are actually shown.*

We added a second legend to the figure.

7. *Fig. 7: The caption does not fit to (a) and (b). I believe (a) and (b) are switched, i.e. (a) is showing the results for the RHI and (b) the results for CLAMPS?*

We corrected this mistake in the new manuscript.

8. *Figs. 7 and 8: In the captions it says "the color scale represent the density of probability of a measurement point" (Fig. 7) and "the probability of occurrence of a measurement point" (Fig. 8). This should be uniform.*

The caption of Fig. 7 is correct and is copied to Fig. 8.

9. *p. 21, l. 25 and Fig. 11: Maybe I am missing it, but I cannot find the information where the RHI profile of dissipation rate come from. Are they averages over same area across the valley or are they individual grid point values?*

They are individual grid points at the same horizontal distance to the WEC as the CLAMPS is located.

10. *p. 25, l. 9: ".. and 0700 UTC (Fig. 12a).*

The reference to the figure is added to the manuscript.

11. *p. 27, l. 1: "...wake induced turbulence being trapped under the inversion..." I don't see enough evidence for this in the data (comment 57 in the first review) and this should be rephrased.*

We rephrase to: "... wake-induced turbulence propagating into the valley with the mean flow "

**2 Relevant changes to the manuscript**

Only the changes described in the review response were made to the manuscript in this minor revision.

**20 References**

Deardorff, J.W.: Sub-Grid-Scale Turbulence Modeling, Advances in Geophysics, 28, 337-343, https://doi.org/10.1016/S0065-2687(08)60193-4, 1985.

[revised manuscript text omitted]

*Copyright statement.* The copyright of the authors Norman Wildmann and Johannes Wagner for this publication are transferred to Deutsches Zentrum fuer Luft- und Raumfahrt e. V., the German Aerospace Center. The copyright of the co-author Julie K. Lundquist is transferred to Alliance for Sustainable Energy, LLC (Alliance) which is the manager and operator of the National Renewable Energy Laboratory (NREL). Employees of Alliance for Sustainable Energy, LLC, under Contract No. DE-AC36-08GO28308 with the U.S. Dept. of Energy, have co-authored this work. The United States Government retains and the publisher, by accepting the article for publication, acknowledges that the United States Government retains a nonexclusive, paid-up, irrevocable, worldwide license to publish or reproduce the published form of this work, for United States Government purposes.

[revised manuscript text omitted]